# Impact of COVID-19 vaccination on symptoms and immune phenotypes in vaccine-naïve individuals with Long COVID

## Abstract

**Background** The symptomatic and immune responses to COVID-19 vaccination of people with Long COVID are poorly characterized.

**Methods** In this prospective study, we evaluated changes in symptoms and immune responses after COVID-19 vaccination in 16 vaccine-naïve individuals with Long COVID. Surveys were administered before vaccination and at 2, 6, and 12 weeks after receiving the first vaccine dose of the primary series. Simultaneously, SARS-CoV-2-reactive TCR enrichment, SARS-CoV-2-specific antibody responses, antibody responses to other viral and self-antigens, and circulating cytokines were quantified before vaccination and at 6 and 12 weeks after vaccination.

**Results** At 12 weeks post-vaccination, self-reported improved health is seen in 10 out of 16 participants, 3 have no change, and 3 have worse health although 2 report transient improvement after vaccination. One participant reporting worse health was hospitalized twice with chest pain (after each dose). Symptom outcomes are most associated with plasma biosignatures. Higher baseline sIL-6R is associated with symptom improvement, and stably elevated levels of IFN-β and CNTF are associated with no improvement. Significant elevation in SARS-CoV-2-specific TCRs and spike protein-specific IgG are observed at 6 and 12 weeks after vaccination. No changes in reactivities are observed against herpes viruses and self-antigens.

**Conclusions** In this study of 16 people with Long COVID, vaccination is associated with increased SARS-CoV-2 spike protein-specific IgG and T cell expansion in most participants. Specific immune features are associated with symptom change after vaccination and most participants experience improved health or no change following vaccination.

## Plain language summary

The impact of the COVID-19 vaccine on unvaccinated individuals suffering from Long COVID is uncertain. This study assessed the experience and biological markers of 16 unvaccinated participants with Long COVID. A total of 10 participants had improved health after vaccination, three reported worsening health, with one hospitalized twice with chest pain. Vaccination boosted the body's immune responses against the virus that causes COVID-19. We identified biological markers that correlate with the changes in overall health after vaccination. Given that the study was small, more research is needed to confirm these results.

---

Long COVID, also known as post-acute sequelae of SARS-CoV-2 infection (PASC), is a debilitating condition following acute SARS-CoV-2 infection[1–6]. It can significantly impact people's lives, including their ability to return to work and engage in other social activities[7,8]. Although investigators have launched several prospective clinical trials of Long COVID treatment[9–12], no definitive therapies exist.

Viral persistence is a possible contributing factor for Long COVID[13–15]. So far, several reports have suggested the persistence of active SARS-CoV-2 reservoirs in Long COVID patients[16,17] and that vaccination could assist in

clearing persistent virus. However, the impact of vaccination after developing Long COVID remains unclear[18]. In a recent study, Nayyerabadi et al. reported the alleviation of symptoms, an increase in WHO-5 well-being scores, and a decrease in inflammatory cytokines after vaccination among participants with Long COVID[19].

At the same time, there are concerns that the vaccine's spike protein or innate immune stimuli induced by the lipid nanoparticles and mRNA may exacerbate Long COVID symptoms by activating immunological pathways[14,20]. These concerns have contributed to vaccine hesitancy among

✉e-mail: akiko.iwasaki@yale.edu

individuals with Long COVID[21]. Consequently, it is crucial to investigate the effect of vaccination on Long COVID symptoms and immunophenotypes by observing individuals before and after their first COVID-19 vaccination, which has remained understudied.

To assess this, we initiated the Yale COVID-19 Recovery Vaccine Study: Measuring Changes in Long Covid Symptoms After Vaccination (NCT04895189), a prospective, unblinded, observational study to evaluate changes in Long COVID symptoms, their prevalence, and burden. Immune responses were evaluated before and after receiving the COVID-19 vaccine to assess vaccine responses and identify factors associated with health outcomes in vaccine-naïve individuals with Long COVID[22]. The study was ended early due to difficulties in recruiting eligible individuals. This report presents findings from 16 participants recruited between May 3, 2021, and February 2, 2022.

Our study shows that at 12 weeks post-vaccination, 10 out of 16 participants reported health improvement, 3 with no change, and 3 with worse health after vaccination in 16 individuals with long COVID. Vaccination is linked to an increase in SARS-CoV-2 spike protein-specific IgG and T cell expansion in the majority of participants. Higher baseline sIL-6R is associated with symptom improvement, while elevated levels of IFN-β and CNTF are associated with no improvement. While future studies are needed to validate these findings, our study offers preliminary indications of biomarkers that may help predict how people with Long COVID respond to vaccines.

## Methods
### Study design
A pre-post, prospective observational study was conducted among unvaccinated individuals experiencing Long COVID symptoms who intended to receive a COVID-19 vaccine as part of routine clinical care (ClinicalTrials.gov Identifier: NCT04895189). Participants completed a survey before vaccination to collect demographic and acute COVID-19 infection information and their baseline (i.e., pre-vaccination) Long COVID symptom experience (survey included in the Supplementary Methods). Participants were vaccinated with any approved COVID-19 vaccine and then asked to complete three follow-up surveys at 2, 6, and 12 weeks after receiving the first vaccine dose of the primary series. SARS-CoV-2 specific humoral responses, responses to common viral pathogens and autoantigens, T-cell repertoire sequencing, and soluble immune modulators were quantified in a subset of participants before vaccination and at 6 (x̄: 6.6) and 12 (x̄: 13.4) weeks after vaccination.

### Ethics statement
This study was approved by the Yale University Institutional Review Board (IRB #2000030423). Informed consent was obtained from all participants at the time of enrollment in the study.

### Patient involvement
Patient advocacy groups were actively engaged in the conception and design of the study. The idea for the study originated from a Survivor Corps poll posted to their Facebook page, many of whom had COVID-19 and suffered from Long COVID. Their poll showed that 40% of respondents with self-reported Long COVID had mild to full symptom resolution after vaccination while 14% reported worsening of their symptoms. In response, hypotheses were developed as to how vaccination might impact Long COVID symptoms[23,24]. Survivor Corps aided in participant recruitment. The Patient-Led Research Collaborative, a self-organized group of Long COVID patient researchers working on patient-led research around the Long COVID experience, was enlisted to contribute to the study design. Both groups advocated for not requiring a positive PCR test for SARS-CoV-2 since access to testing was variable for many people. These groups helped develop the study surveys. Surveys were also informed by prior studies[7,25–29].

### Eligibility
Eligible individuals included unvaccinated individuals 12 years or older who self-reported Long COVID (based on the presence of symptoms that started after COVID-19 and persisted more than two months) and planned to receive the COVID-19 vaccine. To verify past COVID-19 illness, individuals must have had a positive COVID-19 test (PCR or antigen) more than two months prior, have had a positive COVID-19 antibody or T-cell test, had been hospitalized for COVID-19, or had been diagnosed by a clinician as having COVID-19. Participants also had to be willing to travel to New Haven, Connecticut to provide blood and saliva samples. Recruitment was conducted through social media advertisements and patient support groups. Participants were not compensated for their involvement.

### Outcomes
The primary outcome was whether individuals' overall health condition improved, stayed the same, or worsened after receiving a COVID-19 vaccine. Secondary outcomes included changes in symptom prevalence and severity and associated changes in immune response to the COVID-19 vaccine. The immunophenotyping assays included the detection of SARS-CoV-2-specific antibody responses, SARS-CoV-2 specific T cell enrichment, antibody responses to other common viruses and quantitation of soluble immune mediators.

### Data collection
Before vaccination, demographic, acute COVID-19, and persistent symptom information was collected by survey. Participants were asked to rate their symptoms in terms of how much physical pain or discomfort the symptom caused ("physical effects") and how much each symptom impaired their social or family functioning compared to before infection ("social effects") on a 5-level Likert scale from "not at all" to "very much" (Supplementary Table 1). We provided a list of 125 symptoms developed through a literature review and prior Long COVID symptoms lists[7,8,26]. Participants were asked the same questions on the three post-vaccination surveys (2, 6, and 12 weeks after vaccination). Overall health change was measured with the question, "Would you say that your overall health, as compared to your health before the vaccine, is worse, better, or the same?" at 2, 6, and 12 weeks after vaccination. Data collection was performed using RedCap version 12.0.25 (Vanderbilt University). All survey data were self-reported. Blood samples were collected on-site before and at 6 and 12 weeks after vaccination. Further information on the study's design, eligibility criteria, and data collection are available online[22].

### Biospecimen processing
Whole blood was collected in sodium-heparin-coated vacutainers (BD 367874, BD Biosciences) and EDTA-coated vacutainers (BD 367856, BD Biosciences). For each participant, unique study identifiers were provided upon collection. Plasma samples were collected after centrifugation of whole blood at $600 \times g$ for 10 min at room temperature (RT) without brake from sodium-heparin-coated tubes as previously described[30]. The blood samples collected in EDTA-coated tubes were frozen and subsequently shipped to Adaptive Biotechnologies for TCR sequencing.

### Quantitation of SARS-CoV-2 specific antibody levels by ELISA
ELISA assays were performed as previously described[30]. Briefly, MaxiSorp plates (96 wells; 442404, Thermo Scientific) were coated with recombinant SARS-CoV-2 S1 (S1N-C52H3-100 μg, ACROBiosystems), receptor-binding domain (RBD) (SPD-C52H3- 100 μg, ACROBiosystems) and the nucleocapsid protein (NUN-C5227-100 μg, ACROBiosystems) at a concentration of 2 μg/ml in PBS and were incubated overnight at 4 °C. The primary antibodies used for the standard curves were human anti-spike (SARS-CoV-2 human anti-spike [AM006415; 91351, Active Motif]) and human anti-nucleocapsid (SARS-CoV-2 anti-nucleocapsid [1A6; MA5-35941, Active Motif]) and HRP anti-human IgG antibody (1:5,000; A00166, GenScript) was the secondary antibody.

### TCR sequencing & SARS-CoV-2 specific TCR assignment
Immunosequencing of the third complementarity determining (CDR3) regions of TCR-β chains was carried out using ImmunoSEQ Assays

(Adaptive Biotechnologies). Samples were classified as positive or negative for detection and enrichment of COVID-specific T cells using four of Adaptive's COVID-19 classifiers: V1 classifier, V3 classifier, spike classifier and non-spike classifier. The V1 classifier was trained to compare peripheral repertoires from acute COVID and convalescent subjects with control samples collected pre-pandemic[31,32]. The V3 classifier was trained on a larger dataset that included subjects with natural infection as well as those that were vaccinated as positive cases. The sequences in the V3 classifier were cross-referenced against data from MIRA (multiplexed antigen-stimulation experiments) experiment to develop two additional classifier[32,33]. The spike classifier identifies the spike-specific signal, while the non-spike classifier (with vaccinated samples included as controls) identifies natural infection using the non-spike signal. T cell responses are categorized as negative, positive, and "No Call" (representing samples with an insufficient number of T cell rearrangements to make a definitive negative call).

### Rapid extracellular antigen profiling (REAP) library expansion

The new yeast library (Exo205) containing 6452 unique antigens was used. IgG isolations and REAP selections were done as previously described[30]. Briefly, participant IgG was purified from plasma using protein G magnetic beads, and yeast-reactive IgG was initially removed by adsorption to yeast transformed with the pDD003 empty vector. A total of $10^8$ induced Exo205 yeast cells were washed with PBE and incubated with 10 μg of purified participant IgG in duplicate. IgG-bound yeast cells were selected by anti-human IgG Fc antibody binding (clone QA19A42, Biolegend) and next generation sequencing (NGS) was carried out to identify epitopes based on the protein display barcode on yeast plasmids. REAP scores were calculated as described previously[30].

### Multiplex proteomic analysis

Frozen patient plasma was shipped to Eve Technologies (Calgary, Alberta, Canada) on dry ice to run 13 multiplex panels: Human Cytokine/Chemokine 71-plex Discovery Assay (HD71), Human Cytokine P3 Assay (HCYP3-07), Human Cytokine Panel 4 Assay (HCYP4-19), Human Complement Panel Assay (HDCMP1), Human Myokine Assay (HMYOMAG-10), Human Neuropeptide Assay (HNPMAG-05), Human Pituitary Assay (HPTP1), Human Adipokine Panel 2 Assay (HADK2-03), Human Cardiovascular Disease Panel Assay (HDCVD9), Human CVD2 Assay (HCVD2-8), Steroid/Thyroid 6plex Discovery Assay (STTHD) Human Adipokine Assay (HDADK5), and TGF-Beta 3-plex Discovery Assay (TGFβ1-3). Samples were sent in two batches with internal controls in each shipment to assess the effectiveness of batch correction as described below.

To harmonize data across the two batches, ComBat was used, an empirical Bayes method available through the "sva"[34] R package (version 3.4.6), designating the initial batch as the reference and incorporating the following covariates: disease status, sex, age, and hormone conditions. The effectiveness of ComBat was validated using sample replicates between each batch in a matched pairs analysis. Analytes that exhibited significant differences post-correction were excluded from further analysis.

### Statistical analysis

We prospectively sought to enroll 50–100 participants to evaluate overall health and symptom changes. However, the study was terminated early given that few people with Long COVID were vaccine naïve. Our final cohort comprised participants who met eligibility, completed the baseline survey, and were vaccinated at least once.

Cohort characteristics were reported as frequencies with proportions or medians with ranges. The overall health condition of participants before and after vaccination were compared and described as the proportion of individuals with each response (i.e., better, worse, the same, don't know) at each post-vaccination time point out of the number of individuals with a survey submitted at that time point. Baseline collection dates were on an average of 224 (±127) days from the first acute infection symptom, and 255 (±149) days post first infection time point.

For other symptom-related analyses, participants' Likert scale responses to the physical and social effects associated with each symptom were coded numerically (Supplementary Table 1). The distribution of the number of symptoms per participant that resolved, improved, remained not an issue, remained an issue, or worsened stayed the same 2, 6, and 12 weeks after vaccination compared to before vaccination was plotted for the physical and social effects scales, overall and by overall health at 12 weeks (better, the same, worse) (symptom changes are defined in the footnotes of Supplementary Fig. 1). The proportion of participants experiencing each symptom before vaccination was calculated, and for the fifteen most common symptoms before vaccination, the number of participants whose symptom changed—according to the categories above—was tabulated for the physical and social effects scales.

The burden of each participant's symptoms was summarized by summing their responses to the symptom physical and social effect scales separately for each survey. Scores could range from 0 to 500 per survey (i.e., 125 symptoms per survey with a maximum score of 4). Higher values suggest a greater symptom burden, and a value of 0 suggests no symptom burden. Changes in these values indicate a change in the number of symptoms experienced, the symptom severity, or both. We report the median, 2nd and 3rd quartiles, and range for each survey and effect (i.e., physical and social). Differences between surveys were not tested. Missing data were not imputed. Analyses were performed in R (v 4.2.2; R Foundation for Statistical Computing, Vienna, Austria)[35].

Differences in SARS-CoV-2 specific T-cell responses and anti-SARS-CoV-2 antibody responses measured by ELISA and REAP before and after vaccination were assessed using Wilcoxon matched-pairs signed rank tests. To assess the correlation between observed T-cell responses and antibody levels as well as to determine alignment between the two different methods of determining anti-SARS-CoV-2 antibody levels, Spearman rank correlations were calculated. The correlation coefficients between assays were used to measure distances [1-absolute (correlation coefficients)], and hierarchical clustering was conducted using Morpheus[36]. Participants were classified into outcome groups based on self-reported general health status before and after vaccination. The tests were all two-sided and Bonferroni-corrected P-values less than 0.05 were considered statistically significant.

Differences in SARS-CoV-2 T-cell responses, antibody levels, anti-viral antibody levels against common viruses, and autoantibody levels among symptom outcome groups were also compared using Kruskal–Wallis tests. Further, to estimate the average differences in expression of each plasma factor over the course of vaccination, we used linear mixed models via Restricted Maximum Likelihood (REML) regression, estimating the cytokine expression over all three time points amongst three symptom outcome groups: those who did not improve or felt worse at weeks 6 and 12 post-vaccination ($n = 3$; Same/Worse), those who showed transient improvement ($n = 2$, Transient [i.e. Better week 6; then Worse week 12]) and those who reported improvement ($n = 7$, Better). The model incorporated a random effect for each individual as a random intercept, nested within their respective symptom outcome groups. The fixed effects in the model included the symptom outcome and time, along with an interaction term between them to investigate any potential modifying effect of time on the symptom outcome group and adjusted for multiple comparison within each group for these plasma factors using the Tukey method. The analysis was conducted using the JMP statistical software platform (JMP® Pro 17.0.0).

Statistical tests were performed using R (v 4.2.2)[35], GraphPad PRISM(v9.5.1), and JMP statistical software platform (JMP® Pro 17.0.0).

### Machine learning

Unsupervised hierarchical clustering was conducted on 162 plasma-derived analytes obtained from the multiplex proteomic assays to assess patterns of expression across the cohort. Data was standardized by factor and clustering was done based on Ward's distance.

To further identify predictors of symptom improvement from the 162 plasma-derived analytes, we used Partial Least Squares (PLS) analysis via the Non-linear iterative partial least squares (NIPALS) algorithm with k-fold

cross validation (k = 5). All plasma factors and sex were incorporated into the model. Final analysis involved reduction to 4 principal components, which simultaneously minimized the Van der Voet's T-squared statistic (0.00, P = 1.00) and the Root Mean PRESS (0.27) accounting for a sizeable portion of the variance in the data (cumulative pseudo-R-squared = 0.98). Post-analysis, the Variable Importance on Projection (VIP) score was generated for each feature and bootstrapped using Bayesian Bootstrapping. Bias-corrected 95% confidence intervals were calculated. Only features with 95% confidence intervals above the threshold cutoff of 0.8, corresponding to the standard threshold for importance[37,38], were considered significant.

Analysis was conducted using the JMP statistical software platform (JMP® Pro 17.0.0).

### Reporting summary

Further information on research design is available in the Nature Portfolio Reporting Summary linked to this article.

## Results

Among 429 individuals screened between May 3, 2021 and February 2, 2022, 22 met inclusion criteria and consented to participate and 16 individuals completed the baseline survey and received a first dose of a COVID-19 vaccine. Two 2-week surveys and two 6-week surveys were excluded because they were submitted before an earlier survey time point or on the same day as another survey; thus, we included 14 2-week surveys, 14 6-week surveys, and 16 12-week surveys. 14 completed the 2-week post-vaccination survey on time, 14 completed the 6-week survey, and 16 completed the 12-week survey. People not enrolled had already received a vaccine, did not plan to be vaccinated, or could not travel to New Haven for biospecimen collection. The median age of the 16 included participants was 54 years (range 21–69), 13 (81%) were female, and 14 (88%) identified as Non-Hispanic White (Table 1). The median number of months from participants' first onset of symptoms until completing the pre-vaccine survey was 7.2 months (Q1–Q3 4.5–13.9, min-max 2.5–19.6). The median number of months from participants' first self-reported positive COVID-19 test until completing the pre-vaccine survey was 5.5 months (Q1–Q3 4.3–12.9, min-max 1.7–19.5). Two participants did not report positive tests but one reported hospitalization with COVID-19. Immunophenotyping assays were completed on a subset of 11 out of 16 participants because of loss to follow-up in biospecimen collection and instances of difficulty in blood draws where less than expected volumes were collected. All participants reported that they tested positive for COVID-19 at least once with most reporting a PCR-based test (n = 10, 62%). Three (19%) participants were previously hospitalized due to COVID-19 and 4 (25%) visited the hospital or were hospitalized for COVID-19 more than 2 weeks after onset of acute disease (2 of these 4 participants also reported being hospitalized due to COVID-19).

### Pre-vaccination health and symptoms

At baseline, on participants' worst days, 9 (56%) felt they were 50% or less of their health before COVID-19. On participants' best days, 7 (44%) reported feeling 51–75% of their health before COVID-19. The median number of symptoms per participant before vaccination was 23 (Q1–Q3, 13.8–27). The median number of symptoms per participant that resolved before vaccination was 9 (Q1–Q3 5–15, min-max 1–28), and the most common symptoms experienced that had resolved before vaccination were cough, diarrhea, and persistent chest pain or pressure (n = 6 for each).

The fifteen most frequently reported symptoms that had not resolved before vaccination were brain fog (n = 13, 81%), fatigue (12, 75%), difficulty concentrating (11, 69%), difficulty sleeping (10, 63%), heart palpitations (9, 56%), shortness of breath or difficulty breathing (9, 56%), anxiety (8, 50%), memory problems (8, 50%), dizziness (7, 44%), feeling irritable (7, 44%), headache (7, 44%), inability to exercise or be active (7, 44%), nerve sensations (7, 44%), post-exertional malaise (7, 44%).

### Post-vaccination changes in overall health

Eleven of 16 participants (69%) received the Pfizer-BioNTech vaccine (Comirnaty®), 3 (19%) received the Janssen vaccine as their first dose, and 2 (13%) received the Moderna vaccine (SpikeVax®). Nine of 13 participants (69%) recommended to receive a second dose in the primary series reported doing so (i.e., Janssen's vaccine in the primary series was single dose). One participant was hospitalized for chest pain and myocarditis three days after receiving their first vaccine dose and again after their second dose. This participant reported being previously hospitalized soon after infection with probable myocarditis.

Two weeks after vaccination, 6 out of 14 participants with completed surveys reported their health was better (43%), 3 (21%) said their health was the same, 1 (7%) reported worse health, and 4 (29%) were not sure of a change (Fig. 1). At 6 weeks after vaccination, 11 out of 14 (79%) said their health was better than before vaccination, 2 (14%) reported the same health, and 1 (7%) reported worse health. The participant with worse health 2 weeks after vaccination reported better health at 6 weeks. At 12 weeks, 10 out of 16 (62%) reported better health, while 3 (19%) reported the same health and 3 (19%) reported worse health. Two participants who reported better health at 6 weeks reported worse health at 12 weeks, which we classified as transient improvement in subsequent analyses.

The median number of symptoms per participant before vaccination was 23 (Q1–Q3 14.5–27, min-max 3–43, n = 16). In terms of pain or discomfort associated with a symptom, the median (Q1–Q3) number of symptoms per participant that resolved at 2, 6, and 12 weeks was 4.5 (3–7.8), 4 (3–7), and 5 (3–12), respectively, and the number of symptoms per

### Table 1 | Baseline characteristics

| Characteristic | n = 16 (%) |
|---|---|
| Age, median (Min-Max) | 54 (22–69) |
| Missing | 1 |
| Gender | |
| Female | 13 (81%) |
| Male | 3 (19%) |
| Race/ethnicity | |
| American Indian/Alaska Native | 1 (6%) |
| Hispanic | 1 (6%) |
| Non-Hispanic White | 14 (88%) |
| Tested for COVID-19 | 16 (100%) |
| Hospitalized due to COVID-19 | 3 (19%) |
| Hospitalized for COVID-19 or visited a hospital more than 2 weeks after infection | 4 (25%) |
| Test type | |
| Antigen test | 2 (12%) |
| Not sure | 4 (25%) |
| PCR test | 10 (62%) |
| On your best days would you say you are __ of health before COVID-19 | |
| 0–25% of health before COVID-19 | 0 |
| 26–50% of health before COVID-19 | 1 (6%) |
| 51–75% of health before COVID-19 | 7 (44%) |
| 76–100% of health before COVID-19 | 8 (50%) |
| On your worst days would you say you are __ of health before COVID-19 | |
| 0–25% of health before COVID-19 | 5 (31%) |
| 26–50% of health before COVID-19 | 4 (25%) |
| 51–75% of health before COVID-19 | 6 (38%) |
| 76–100% of health before COVID-19 | 1 (6%) |

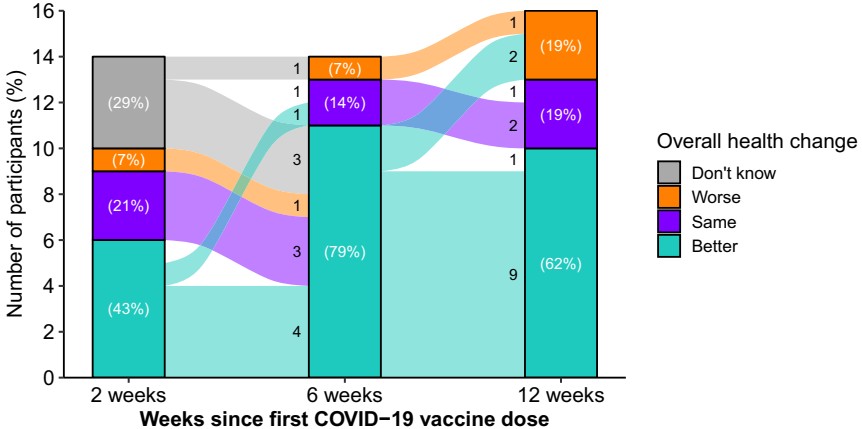

Fig. 1 | Overall health change since receiving first dose of COVID-19 vaccine, measured with surveys sent 2, 6, and 12 weeks after vaccination. Participants were asked "Would you say that your overall health, as compared to your health before the vaccine, is worse, better, or the same?" at each post-vaccination survey. Data missing for $n = 2$ at 2 weeks and $n = 2$ at 6 weeks.

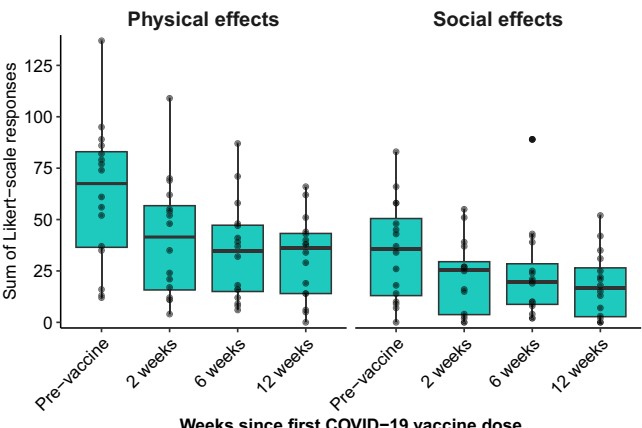

Fig. 2 | Distribution of the sum of participants' responses to two measures of symptom severity—physical and social effects—measured before vaccination and surveys sent 2, 6, and 12 weeks after vaccination. To measure physical effect of each symptom from a list of 125 symptoms, participants were asked, "While experiencing these symptoms, how much do/did they bother you in terms of discomfort or pain?" Similarly, to measure social effects, participants were asked, "After quarantine, how much does/did the symptom impair your social or family functioning compared to pre-COVID? Responses for each symptom were scored 0–4 (Supplementary Table 1) and summed for each participant. Boxplots show the distribution of responses, with points indicating the score for each participant; the central lines indicate the group median values, the top and bottom lines indicate the 75th and 25th percentiles, respectively, the whiskers represent 1.5× the interquartile range. Higher values suggest a greater symptom burden, and a value of 0 suggests no symptom burden Data missing for $n = 2$ at 2 weeks and $n = 2$ at 6 weeks.

participant that improved but did not fully resolve was 6 (6–11), 9.5 (8.3–13.3), and 11 (5–13), respectively (Supplementary Fig. 1). The median (Q1–Q3) number of symptoms per participant that worsened at 2, 6, and 12 weeks was 3 (2.5–5), 3 (2.5–5), 3 (1–4.3), respectively. Similar trends were observed regarding impairment of social and family functioning associated with a symptom, though with a higher number of symptoms per participant resolving. Symptom changes by overall health at 12 weeks is shown in Supplementary Fig. 2. For the 15 most common symptoms experienced before vaccination, the number of participants whose symptoms resolved, improved, remained not an issue, remained an issue, or worsened are presented in Supplementary Data 1 and 2. The median number of symptoms that bothered participants "very much", "quite a bit", "somewhat", "a little bit", or "not at all" before and after vaccination are provided in Supplementary Table 2.

Symptom burden appeared to decrease after vaccination on both physical and social effect scales (Fig. 2). Before vaccination, the

median physical effect score for all symptoms was 63 (Q1–Q3 37–83, min-max 12–137, $n = 16$) and the median social effect score was 35 (Q1–Q3 13–51, min-max 0–83, $n = 16$), where higher values represent worse symptom burden. Compared with before vaccination, the median physical effect score decreased to 42 (Q1–Q3 18–60, min-max 4–109, $n = 14$) at 2 weeks after the first COVID-19 vaccine dose, then 35 (Q1–Q3 16–46, min-max 8–87, $n = 14$) 6 weeks after vaccination, and 36 (Q1–Q3 14–43, min-max, 0–66, $n = 16$) 12 weeks after vaccination. At 2 weeks after the first COVID-19 vaccine dose, the median social effect score decreased to 26 (Q1–Q3 6–27, min-max 0–55, $n = 14$), then 21 (Q1–Q3 9–36, min-max 2–89, $n = 14$) 6 weeks after vaccination, and 17 (Q1–Q3 3–27, min-max 0–52, $n = 16$) 12 weeks after vaccination.

### SARS-CoV-2-specifc T-cells and antibody responses

To characterize the T-cell responses to SARS-CoV-2, sequencing of the CDR3 regions of T-cell receptor-β (TCR- β) chains was carried out. There was a significant increase in spike protein (Fig. 3a; $P = 0.012$, V1(Fig. 3b; $P = 0.011$) and V3 (Fig. 3c; $P = 0.011$) classifier scores at 6 weeks post-vaccination, which was indicative of an increase in SARS-CoV-2 specific T-cell clonal depth and breadth. By contrast and as expected, no significant differences were observed in classifier scores for non-spike protein TCRs with vaccination (Fig. 3d). There were some individuals who retained high SARS-CoV-2 specific TCR clonality at 12 weeks post-vaccination, however the differences in model scores were not statistically significant in comparison with pre-vaccination. There was a significant decrease in V3 classifier score at 12 weeks post-vaccination as compared to 6 weeks ($P = 0.011$), however this observation was not replicated using the V1 or spike specific classifier scores.

Next, SARS-CoV-2 antibody responses were evaluated. A significant increase in anti-S1 IgG (Fig. 3e; pre vs 6 weeks: $P = 0.003$, pre vs 12 weeks: $P = 0.003$) and anti-RBD IgG (Fig. 3f; pre vs 6 weeks: $P = 0.003$, pre vs 12 weeks: $P = 0.003$) levels at 6 weeks and 12 weeks post-vaccination was observed without significant rise in anti-N IgG levels (Fig. 3g). The anti-S1 and anti-RBD IgG antibody levels peaked at 6 weeks (median anti-S1 IgG: $8.8 \times 10^4$ ng/mL; median anti-RBD IgG: $5.0 \times 10^5$ ng/mL) with a marginal decrease at 12 weeks (median anti-S1 IgG: $5.8 \times 10^4$ ng/mL; median anti-RBD IgG: $2.2 \times 10^5$ ng/mL). To further validate the humoral responses attributed to vaccination, SARS-CoV-2 spike protein reactivities were assessed using REAP. Participant antibody reactivities against Beta, Delta, Epsilon, and Omicron variant RBD epitopes were independently evaluated. A significant increase in reactivity across all non-Omicron RBD epitopes at 6 weeks post-vaccination (Supplementary Fig. 3a–c) and the Epsilon variant across 6 and 12 weeks post-vaccination (Fig. 3h; pre vs 6 weeks: $P = 0.011$, pre vs 12 weeks: $P = 0.023$) was observed.

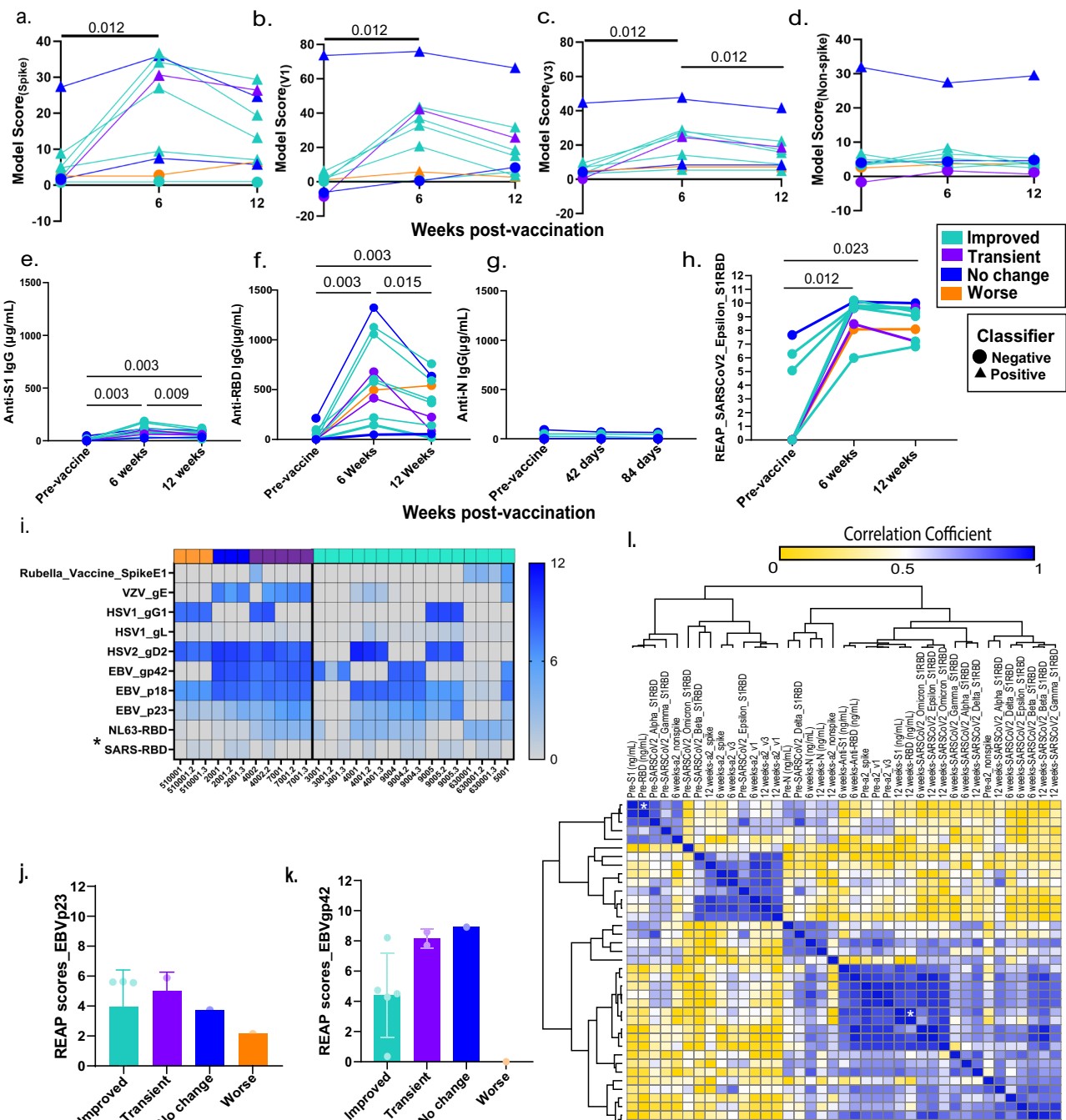

**Fig. 3 | Vaccination resulted in increase in SARS-CoV-2 T-cell repertoires and specific humoral responses among Long COVID participants. a** Model scores and binary classifications are plotted against days post-vaccination using spike viral protein specific classifier, **b** COVID classifier version 1 (v1), **c** COVID classifier version 3 (v3), **d** Non-spike-specific protein classifier, **e** Line plots of matched anti-SARS-CoV-2 S1 IgG concentrations before, 6 and 12 weeks post-vaccination in Long COVID participants, **f** Line plots of matched anti-SARS-CoV-2 RBD IgG concentrations before, 6 and 12 weeks post-vaccination in Long COVID participants, **g** Line plots of matched anti-SARS-CoV-2 N IgG concentrations before, 6 and 12 weeks post-vaccination in Long COVID participants. The color codes denote the reported health status at 6 and 12 weeks post-vaccination, better at both timepoints [teal], no change at both timepoints [blue], better at 6 weeks and worse at 12 weeks, [purple] & worse at both timepoints [orange], **h** Line plots of matched anti-SARS-CoV-2 Epsilon variant reactivity scores against the Spike protein assessed by Rapid

Extracellular Antigen Profiling (REAP), **i** Heatmap of REAP reactivities against 10 viral proteins namely, proteins belonging to common viral pathogens from *Coronaviridae* (human SARS-CoV-1 viruses), *Herpesviridae* families, and the Rubella vaccine protein. Each protein and each participant timepoint are represented as a row and a column respectively. The participant IDs are mentioned below each column and the numbers after decimal denote the collection timepoints after vaccination (6 weeks = 2; 12 weeks = 3). Statistical significance determined by Wilcoxon Rank tests and corrected for multiple testing using the Bonferroni method, **j** EBV p23 REAP scores among outcome groups. Significance was assessed using Kruskal–Wallis tests, **k** EBV gp42 REAP scores among outcome groups, **l** Hierarchical clustering of Spearman Rank correlation coefficients of TCR model scores, antibody concentrations and REAP scores at all three timepoints. Only adjusted *p*-values of <0.05 are mentioned in line plots and denoted by asterisks in heatmaps.

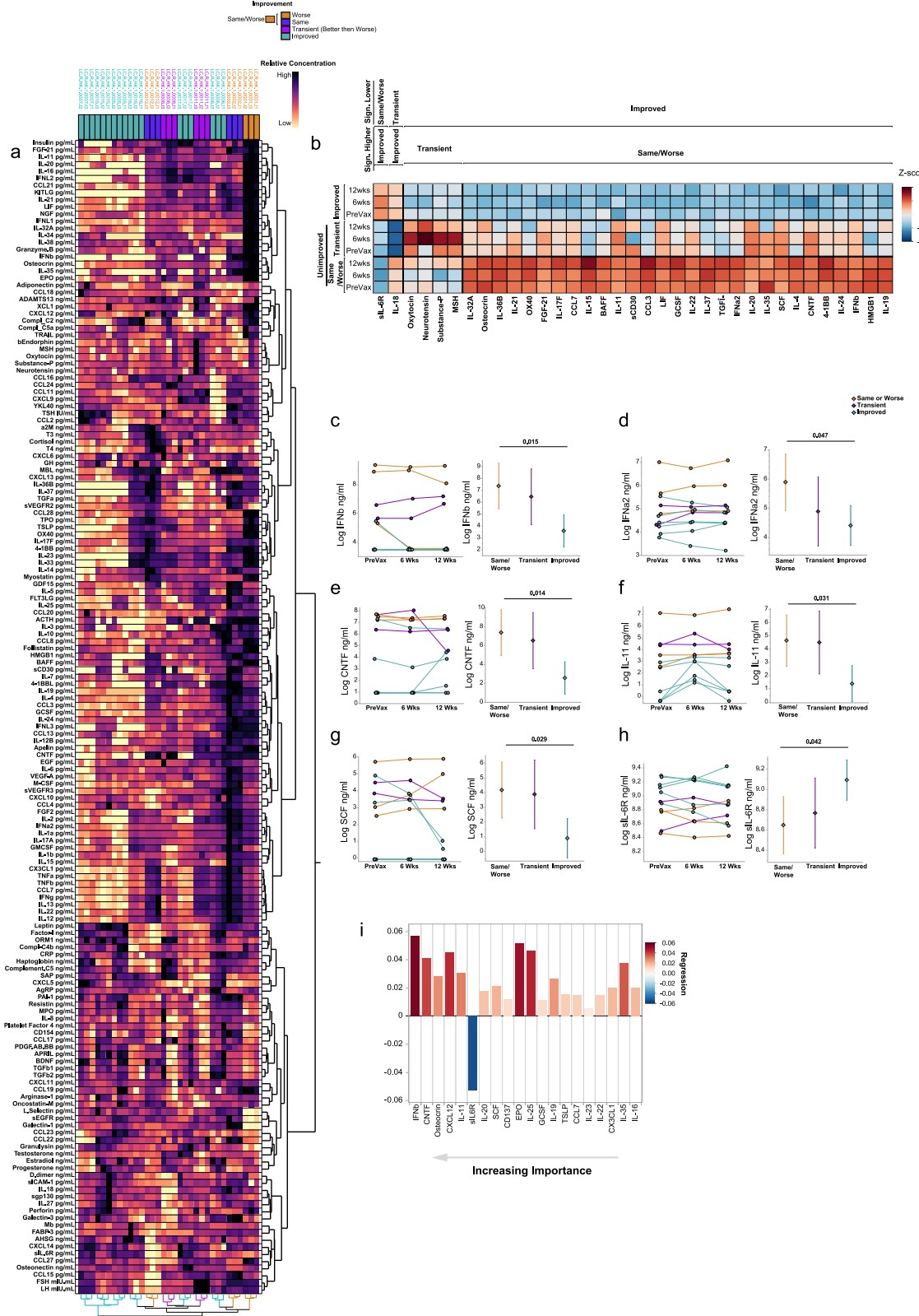

## IgG responses to herpesviruses and autoantibodies to the extracellular proteome

Given that latent virus reactivation has been a hypothesis behind Long COVID pathobiology and evidence of recent Epstein-Barr Virus (EBV) reactivation has been reported[30,39,40], anti-viral REAP reactivities against two families of common viral pathogens namely, *Coronaviridae* (human SARS-

CoV-1 viruses) and *Herpesviridae*, were assessed. Rubella vaccine spike antigen served as internal control as no changes were expected in reactivities with COVID-19 vaccination. As expected, there was a significant increase in REAP scores against SARS-CoV-1 RBD upon vaccination at 6 weeks (Fig. 3i; $P = 0.047$). This increase was maintained at 12 weeks, despite not being statistically significant after multiple testing correction ($P = 0.09$).

**Fig. 4 | Elevated interferon and neuropeptide signaling is associated with poor recovery post-vaccination. a** Unsupervised hierarchical clustering of plasma-derived analyte expression within the cohort for all three sample timepoints (pre-vaccination, 6 weeks post series completion, and 12 weeks post series completion). Color panel above heatmap shows the symptom outcome subgroup of each individual as indicated by the key. Samples for each individual are labeled by their sample code LC.R.HK.1.00XX.tX, where XX designates the patient ID and tX designates the timepoint (t1 = pre vaccination, t2 = 6 weeks post series completion, and t3 = 12 weeks post series completion). Sample label color indicates further categorization into Same/Worse (orange), Transient improvement (i.e., better then worse; purple), and Improved (teal). Color scale is magma and is normalized for each analyte (data table columns) with darker colors indicating higher relative expression and lighter colors indicating lower expression as shown by the key. **b** Expression Heatmap of significant differentially expressed factors between symptom outcome groups (Same/Worse, Transient, and Improved), as labeled. Each subgroup is further separated by the vaccine timepoint. Each factor was centered and standardized to generate a z-score and colors are representative of expression as indicated by the legend (positive z-scores in red; negative z-scores in blue). To show significance between groups, samples were organized with outer brackets of the heatmap indicating the symptom outcome group demonstrating significantly lower expression and inner brackets indicating the comparator group from which significance is derived. Significance was determined using linear mixed models (LMM) via restricted maximum likelihood (REML) regression for log-transformed values, accounting for repeated measures across individuals over time as described in the methods and adjusted for multiple comparisons within each parameter using the Tukey method. **c–h** Example differentially expressed factors between symptom outcome groups as determined by LMM, previously described. **i** Top 20 bootstrapped predictors of symptom outcome (unimproved vs improved), determined by Partial Least Squares (PLS) optimized at eight components. Predictors are ordered by importance with highest importance on the left. Color and direction of each bar represents the relative regression association to unimproved individuals with positive values showing a positive association (red) and negative values showing a negative association (blue). Color is determined by regression as shown. Details of NIPALS and detailed results can be found in the methods and in extended data, respectively.

Herpesvirus reactivities varied across participants. However, no significant decrease in reactivities was observed post-vaccination among the herpesvirus antigens tested including EBV (Fig. 3i; Supplementary Data 3). Additionally, no differences in median reactivities were observed against EBV proteins p23 and gp42 across outcome groups at 6 and 12 weeks post-vaccination (Fig. 3j, k).

Next, given prior reports of elevated autoantibodies targeting the exoproteome in severe acute COVID-19[41], we assessed for changes in extracellularly targeted autoantibodies during vaccination (Supplementary Fig. 4a). No difference in the number of autoantibody reactivities at baseline (Supplementary Fig. 4b) or in the mean REAP score delta, representing the change in autoantibody magnitude over time (Supplementary Fig. 4c), between the groups was observed. Overall, autoantibodies were stable over time during vaccination (Supplementary Fig. 4c–e), with the mean REAP score delta close to 0 for all groups. These results are in alignment with a previous report focusing on autoantibody dynamics during SARS-CoV-2 mRNA vaccination in healthy individuals without Long COVID[42].

### Correlation between SARS-CoV-2 specific TCR and antibody levels

To further evaluate the relation between SARS-CoV-2 specific TCR scores with antibody levels and to assess the concordance among the orthogonal methods of antibody detection, correlation analyses were carried out. Three distinct clusters emerged when distances were calculated based on correlation values among TCR classifier scores and anti-SARS-CoV-2 antibody concentration as well as between ELISA and REAP assays at different timepoints. Each cluster indicated that there was a general concordance in antibody levels using orthogonal methods and TCR scores based on Spearman's r ($r_s$) and unadjusted p-values (Fig. 3l, Supplementary Data 4 and 5). It was also observed that higher numbers of pre-vaccination SARS-CoV-2 specific TCR repertoire resulted in higher titers of antibodies both at pre-vaccination, 6- and 12-weeks post-vaccination along with an increase in spike protein specific TCR repertoire. Despite visually strong correlation patterns, due to the small sample size, only anti-SARS-CoV-2 S1 and anti-RBD antibody levels as detected by ELISA at pre-vaccination timepoint and at 12 weeks were statistically significant after multiple testing corrections (pre-vaccination: $r_s = 0.96$, $P = 0.021$; 12 weeks post-vaccination: $r_s = 0.98$, $P = 0.003$; Supplementary Data 6).

No significant differences were observed between post-vaccination increase in SARS-CoV-2 specific TCR classifier scores and improvement in overall health status. Similarly, no differences were also observed in self-reported health status and increase in anti-SARS-CoV-2 antibody.

### Soluble immune mediators

To understand the impact of vaccination on the cytokine, hormone, and proteomic profiles of individuals with Long COVID, unsupervised

hierarchical clustering of 162 analytes measured in their plasma was first conducted (Fig. 4a). Clustering analysis showed a consistent pattern in their plasma expression profiles at 6- and 12-weeks post-vaccination. Samples clustered by individual and not by timepoint post-vaccination, suggesting an entrenchment in the cytokine profile of each individual that was not significantly affected by vaccination.

To understand the relationship of these plasma-derived analytes with post-vaccine symptom outcomes, the average expression levels of each analyte was compared over all three timepoints amongst three symptom outcome groups: those who did not improve or felt worse at weeks 6 and 12 post vaccination ($n = 3$; Same/Worse), those who showed transient improvement ($n = 2$, Transient [i.e. Better week 6; then Worse week 12]) and those who reported improvement ($n = 7$, Better). To do so a linear mixed model was constructed using restricted maximum likelihood (REML) regression for each cytokine and accounted for both time and the interaction of time with each outcome group. Thirty-five factors were found to be significant amongst these subgroups, with the majority being significantly elevated in the Same/Worse group compared with the Improved group. Type I interferons were higher at baseline and after vaccination amongst the Same/Worse group, including IFN-β and IFN-α, compared to the Improved group (Fig. 4b–d). Ciliary neurotrophic factor (CTNF; a neuropeptide that is released by the hypothalamus), IL-11, and stem cell factor (SCF) were also significantly elevated in the Same/Worse group compared to the Improved group (Fig. 4e). Other neuropeptides were elevated amongst the transient group including oxytocin, neurotensin, substance P and melanocyte stimulating hormone (MSH) (Fig. 4b). Notably, soluble IL-6 receptor (sIL-6R; an anti-inflammatory protein responsible for mitigating IL-6 signaling), was significantly higher amongst those who showed improvement compared to the Same/Worse group (Fig. 4h).

Further Partial Least Squares (PLS) analysis with 5-fold cross validation was employed on all 162 analytes to determine feature importance as predictors of symptom outcome and evaluate concordance with the significant features obtained from our LMM models. Final analysis involved reduction to 8 components, accounting for a sizeable portion of the variance in the data (cumulative pseudo-R-squared = 0.99). The top two significant predictors of the PLS analysis for non-improvement were IFN-β and ciliary neurotrophic factor (CNTF) respectively (Fig. 4i). The top significant predictor of improvement was sIL-6R (Fig. 4i), while sgp130, an important immunological partner to sIL-6R, was also associated with improvement, passing the initial VIP threshold criteria (Supplementary Fig. 5), though not the additional bootstrapping threshold criteria. Taken together these results suggested that high IFN and neuropeptide signaling were predictors of non-improvement while those involved in mitigating cytokine signaling, namely sIL-6R, was a predictor of improvement.

## Discussion

In this prospective cohort study of 16 vaccination-naïve individuals with Long COVID and extensive symptoms at baseline, it was observed that most people improved or stayed the same during follow-up, but some experienced worsening, including one participant who was hospitalized with chest pain and myocarditis. These outcomes were associated with their plasma-derived biosignatures, suggesting that these immune signatures may serve to differentiate or predict outcomes in future larger studies. However, it is important to note that this study lacked concurrent controls, was unblinded, and small, so it is challenging to make definitive statements about the effect of vaccination or these identified immune signatures, particularly since many people with Long COVID have fluctuations in their symptoms. Challenges in recruiting individuals who met eligibility criteria limited the sample size to 16 people, 32% of its target size. Future studies with controls are needed to understand the effect of vaccination on Long COVID symptoms.

Some studies and systematic reviews have reported improvement or non-significant change in self-reported health among people with Long COVID who were vaccinated for the first time[18,43–45]. Our findings are consistent with these reports. A single-center observational study in the United Kingdom identified 44 Long COVID patients (reporting a median of 4.1 and 3.6 symptoms per patient) who had received at least one dose of a COVID-19 vaccine and interviewed at 1 month and 8 months post-vaccination with the SF-36 and Warwick and Edinburgh Mental Wellbeing scores[46]. After adjustment, health status measured with these instruments at 8 months did not differ compared to Long COVID patients who were not vaccinated. In an online cross-sectional survey study of 2094 people in Switzerland, 35.5% of participants reported that their Long COVID symptoms improved, 28.7% reported their symptoms were stable, and 3.3% reported their symptoms worsened after vaccination[47]. In a French target trial emulation study from the ComPaRe Long COVID cohort, COVID-19 vaccination was associated with reduced Long COVID severity and symptom burden at 120 days compared with those unvaccinated[48].

Possible mechanisms of Long COVID have been proposed as: 1) a persistent viral reservoir or "viral ghost," which are fragments of the virus (RNA, proteins) that linger after the infection has been cleared but are still capable of stimulating the immune system; 2) an autoimmune response induced by the infection; 3) reactivation of latent viruses; and 4) tissue dysfunction that results from inflammation triggered by the infection[14,20]. Under these hypotheses, COVID vaccination may alleviate Long COVID symptoms through vaccine-induced T cells and antibody responses that may be able to eliminate the viral reservoir, and the "viral ghost," diversion of autoreactive leukocytes, or removal of inflammatory sources leading to tissue dysfunction. Vaccination could also theoretically indirectly contribute to controlling reactivated latent viruses by restoring proper T and B cell immunity against these herpesviruses indirectly through elimination of SARS-CoV-2 infected cells and antigens[49].

Although we have a small sample size, this study provides evidence for alleviating symptoms among Long COVID participants upon vaccination, along with an expected increase in SARS-CoV-2 specific T-cell repertoire and anti-SARS-CoV-2 spike protein specific IgG levels. However, clear results of hypothesis 1 (persistent viral reservoir) testing will be available once the results of the Paxlovid trials (NCT05595369[50], NCT05668091[10], NCT05823896[51], NCT05576662[11]) and monoclonal anti-spike antibody (NCT05877508) are shared with the scientific community. Thus far, two clinical trials have failed to demonstrate health improvement after a 15 day course of Paxlovid treatment in long COVID[52–54]. In addition, a recent study did not find evidence of changes in circulating viral proteins in response to vaccination in those with Long COVID[19].

Our study found that the plasma-derived soluble analyte profile showed a very stable pattern before and after vaccination, suggesting that vaccines had a minimal effect on the cytokine dynamics of individuals at least at the time points measured. Without concurrent controls, it is difficult to assess if this phenomenon is unique to individuals with Long COVID or is similar in controls. Nevertheless, an overall elevated cytokine pattern—namely in interferon and neuropeptide signaling— was detected among those who did not improve or showed only transient symptomatic improvement post-vaccination. These findings pose an interesting observation that may help identify predictors of improvement versus non-improvement in larger studies. Elevated interferon signaling could suggest the possibility of an ongoing infectious viral process in these individuals or sustained inflammatory condition triggered by the acute infection. The lack of improvement post-vaccination and the persistence of this signaling could suggest that either the vaccine was incapable of producing the necessary antibodies and T cells that clear persistent infection when a viral reservoir exists, or that the main driver of disease in such individuals is not SARS-CoV-2, but re-emergence of a latent infection such as EBV or autoimmunity. It is also possible that the elevated IFN signaling pattern may be a sign of persistent immune dysregulation, and not from ongoing infection. More work will be needed to confirm the findings of this small study and, in turn, decipher a possible mechanism for the elevation of interferon in these individuals, including the exploration of CNS involvement due to the elevation of neuropeptides, which were also associated with poor improvement. A recent study has shown that the persistence of IFN signaling can lead to lower serotonin levels, a critical neurotransmitter[55] that may also be involved in the symptom profile of individuals and or these outcomes. However, given the small sample size of our study, these possibilities can only be interpreted as speculative.

The limitations of this study include the lack of concurrent controls, which we did not include for ethical reasons, the small sample size, and our inability to recruit more participants. These factors limit the ability to estimate the benefit or harm caused by vaccination. Symptom and immune changes may be a result of the natural course of disease not due to the vaccine. Ideally, this study should have been conducted when there was more equipoise about the benefits of vaccination such that vaccinated participants would be more comparable to unvaccinated individuals with respect to the risk of Long COVID outcomes. Moreover, we did not have much diversity in this small cohort. Participants had to be physically able to travel, so they may have been less likely to have severe Long COVID; at the same time, individuals more severely affected by Long COVID may have been more motivated to meet the travel requirements to participate in the study, as well as to be vaccinated. Participants also had to have the financial means and occupational flexibility to travel. Additionally, generalizability to all individuals with Long COVID cannot be determined, especially for those who have developed Long COVID symptoms later in the pandemic (e.g., post-Omicron era). Participants procured their vaccinations and received vaccines from three different manufacturers. Potential differences in symptoms and immunophenotype changes between manufacturers were not examined due to the sample size. Surveys from three participants were excluded from the analysis because their submissions were out of order or on the same day. Among these participants at 12 weeks, one reported overall better health and two reported the same health compared to before vaccination. It's possible the timing of survey submissions was related to disease severity, for instance delaying study surveys until symptoms subsided, but this is unknown. Finally, surveys were not designed to systematically collect information on new symptoms that began after vaccination; only symptoms present before vaccination were queried after vaccination. However, this study's strengths are the prospective study design of vaccine naïve individuals with Long COVID, an increasingly rare population, with assessment of symptom burden, degree of physical and social disability, and immunophenotyping at multiple time points after vaccination.

In conclusion, in this study of 16 individuals living with Long COVID, most people improved or stayed the same, though some had worsening symptoms. Vaccination resulted in an increase in SARS-CoV-2 specific T-cell populations and anti-spike protein IgG levels. The top predictors of participant non-improvement upon vaccination were stable elevated levels of IFN-β and CNTF. Elevated levels of sIL-6R were found to be a predictor of improvement. Future studies are needed to better understand the impact of vaccination on the health of people living with Long COVID.

## Data availability

All data supporting the findings of this study are available within the paper and its Supplementary Information (including Supplementary Data 1–10). Raw data are available from the corresponding author on reasonable request.

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

## Acknowledgements
We want to thank the participants who contributed to the study and patient groups for their recruitment efforts and study design contributions. This work was supported in part by funds from Fred Cohen and Carolyn Klebanoff and by grants from National Institute of Allergy and Infectious Diseases (R01AI157488 to A.I.), the Howard Hughes Medical Institute Collaborative COVID-19 Initiative (to A.I.), and the Howard Hughes Medical Institute (to A.I.). J.J., T.T., and B.B. received research support from Yale University from the Food and Drug Administration for the Yale-Mayo Clinic Center of Excellence in Regulatory Science and Innovation (CERSI) (U01FD005938).

## Author contributions
C.B.G. Methodology, Formal analysis, Writing – Original Draft, Writing – Review & Editing, Visualization; B.B. Methodology, Formal analysis, Data Curation, Writing – Original Draft, Writing – Review & Editing, Visualization; J.S. Methodology, Formal analysis, Writing – Original Draft, Writing – Review & Editing, Visualization; J.J. Methodology, Formal analysis, Visualization, Writing – Review & Editing; LWL Writing – Review & Editing, V.S.M. Methodology, Writing – Review & Editing; M.S. Methodology, Writing – Review & Editing; D.M. Conceptualization, Data Curation, Writing – Review & Editing; C.C. Conceptualization, Data Curation, Writing – Review & Editing; J.R.G. Writing – Review & Editing; A.T. Writing – Review & Editing; T.M. Writing – Review & Editing; C.L. Writing – Review & Editing; M.A.P. Writing – Review & Editing; L.X. Writing – Review & Editing; T.J.T. Writing – Review & Editing; T.T. Writing – Review & Editing; J.H. Methodology Writing – Review & Editing; D.B.G. Conceptualization, Methodology, Writing – Review & Editing; A.A. Conceptualization, Methodology, Writing – Review & Editing; G.A. Conceptualization, Methodology, Writing – Review & Editing; H.D. Conceptualization, Methodology, Writing – Review & Editing; K.H. Conceptualization, Methodology, Writing – Review & Editing; L.M. Conceptualization, Methodology, Writing – Review & Editing; W.L.S. Conceptualization, Writing – Review & Editing; D.G. Writing – Review & Editing; H.W. Conceptualization, Writing – Review & Editing; A.M.R. Conceptualization, Writing – Review & Editing; L.G. Methodology, Writing – Review & Editing; C.D.C. Writing – Review & Editing; A.I. Conceptualization, Methodology, Resources, Writing – Original Draft, Writing – Review & Editing, Supervision, Project Administration, Funding Acquisition; H.M.K. Conceptualization, Methodology, Resources, Writing – Original Draft, Writing – Review & Editing, Supervision, Project Administration, Funding Acquisition.

## Competing interests
In the past three years, H.M.K. received expenses and/or personal fees from United Health, Element Science, Eyedentifeye and F-Prime; he is a co-founder of Refactor Health and HugoHealth; and is associated with contracts, through Yale New Haven Hospital, from the Centers for Medicare & Medicaid Services and through Yale University from the Food and Drug Administration, Johnson & Johnson, Google and Pfizer. A.I. co-founded and consults for RIGImmune, Xanadu Bio and PanV and is a member of the Board of Directors of Roche Holding and Genentech. No other authors declare competing interests.

## Additional information

Connor B. Grady [1,2,19], Bornali Bhattacharjee [3,4,19], Julio Silva [3,19], Jillian Jaycox[3], Lik Wee Lee[5], Valter Silva Monteiro [3], Mitsuaki Sawano [6], Daisy Massey[6], César Caraballo[6,7], Jeff R. Gehlhausen[3], Alexandra Tabachnikova[3], Tianyang Mao [3], Carolina Lucas [3], Mario A. Peña-Hernandez[3,8], Lan Xu[3], Tiffany J. Tzeng[3], Takehiro Takahashi[3], Jeph Herrin[7], Diana Berrent Güthe[9], Athena Akrami[10,11], Gina Assaf[11], Hannah Davis[11], Karen Harris[9], Lisa McCorkell[11], Wade L. Schulz[4,6,12], Daniel Griffin [13], Hannah Wei[11], Aaron M. Ring[3], Leying Guan [4,14], Charles Dela Cruz[4,8,15], Harlan M. Krumholz[4,6,17,18,20] & Akiko Iwasaki [3,4,16,20] ✉

[1]Department of Biostatistics, Epidemiology, and Informatics, Perelman School of Medicine, University of Pennsylvania, Philadelphia, PA, USA. [2]Department of Epidemiology, Harvard T.H. Chan School of Public Health, Boston, MA, USA. [3]Department of Immunobiology, Yale University School of Medicine, New Haven, CT, USA. [4]Center for Infection and Immunity, Yale School of Medicine, New Haven, CT, USA. [5]Adaptive Biotechnologies, Seattle, WA, USA. [6]Center for Outcomes Research and Evaluation, Yale New Haven Hospital, New Haven, CT, USA. [7]Department of Internal Medicine, Yale School of Medicine, New Haven, CT, USA. [8]Department of Microbial Pathogenesis, Yale University School of Medicine, New Haven, CT, USA. [9]Survivor Corps, Washington, DC, USA. [10]Sainsbury Wellcome Centre, University College London, London, UK. [11]Patient-Led Research Collaborative, Washington, DC, USA. [12]Department of Laboratory Medicine, Yale University School of Medicine, New Haven, CT, USA. [13]Department of Medicine, Columbia University Vagelos College of Physicians and Surgeons, New York City, NY, USA. [14]Department of Biostatistics, Yale School of Public Health, New Haven, CT, USA. [15]Department of Medicine, Section of Pulmonary and Critical Care Medicine, Yale University School of Medicine, New Haven, CT, USA. [16]Howard Hughes Medical Institute, Chevy Chase, MD, USA. [17]Department of Health Policy and Management, Yale School of Public Health, New Haven, CT, USA. [18]Section of Cardiovascular Medicine, Department of Internal Medicine, Yale School of Medicine, New Haven, CT, USA. [19]These authors contributed equally: Connor B. Grady, Bornali Bhattacharjee, Julio Silva. [20]These authors jointly supervised this work: Akiko Iwasaki, Harlan M. Krumholz. ✉e-mail: akiko.iwasaki@yale.edu

