## [Transparent Peer Review file · Communications Medicine]

Impact of COVID-19 vaccination on symptoms and immune phenotypes in vaccine-naïve individuals with Long COVID

Corresponding Author: Professor Akiko Iwasaki

Version 0:

Reviewer comments:

Reviewer #1

(Remarks to the Author)

This manuscript reports a small but interesting study on 16 people with Long Covid in terms of the effect of 1 vaccine dose on symptoms, spike Ab and T cell immunity, a REAP Ab array panel, and Luminex cytokine responses.

For the most part, its too small a study for major answers, but there are nevertheless some interesting components.

Much of my reviewer feedback to the authors relates to nuance in terms of how they have elected to report the results : On reaching the Discussion, the authors' message is that the study may allay the vaccine-hesitancy of some with Long Covid fearing that vaccination may exacerbate their symptoms. However, this point - an important one - is never highlighted in the Abstract. Even in the Discussion, a few extra sentences would be helpful in relation to data from Ziyad Al-Aly and others on the benefit of vaccination against ongoing/cumulative LC risk from repeat infections.

There's a real glass-half-full or half-empty dilemma in how/whether to report 'symptom improvement':in the 'improved' group, the median number of symptoms had decreased from 23 to 15.5. At first sight, this doesn't seem a huge improvement. Is there any way the authors can better annotate the improvement or otherwise to quality of life? ie was this a substantive improvement?

Perhaps the biggest weakness, apart from the lack of an unvaccinated control group, is that the study only proceeds as far as week 12. Since its only 16 people, is there any way of going back to them and asking whether improvement was sustained at 6 or 12 months?

Another weakness is that the study encompasses three different vaccines.

Sixteen is a very small number of individuals to stratify into improved versus not-improved, but nevertheless, the biosignature they get from the plasma analytes is probably the most newsworthy part of the paper - lines 474-475 - but this didn't come through very clearly in the Abstract or Discussion.

Reviewer #2

(Remarks to the Author)

Manuscript Number: COMMSMED-24-0059

Impact of COVID-19 vaccination on symptoms and immune phenotypes in vaccine-naïve individuals with Long COVID

Thank you for the opportunity to review the manuscript titled, "Impact of COVID-19 vaccination on symptoms and immune phenotypes in vaccine-naïve individuals with Long COVID." While an interesting premise, based on the cited literature it seems like this question has already been addressed. Unfortunately, I do not see that the present study contributes to the literature, especially as the sample size of this study is too small to even attempt to draw statistical conclusions.

I have provided specific comments and questions below.

Methods:

1. How long before the study were people's infections? That is not mentioned and would impact the prevalence and severity of long COVID symptoms, according to reports in the current literature.

a. There was also no mention of how long people had been experiencing symptoms, and whether they had been improving, stable, or worsening prior to the study's vaccine intervention.

2. There does not appear to be any attempt at having a control group, although there are several comparisons that could have been done. Ideally, you would have a sham vaccine group (or a group who received a vitamin, or a different vaccine,

etc.) so that you could compare changes in symptoms and their severity (S&S). You also could have compared S&S to individuals with long COVID who HAD been vaccinated to help understand the impact vaccination might have had on symptom progression.

3. Why were TCR and immune markers only quantified in a subset? Who were these individuals?

4. I am not a fan of combining symptoms and their severity into one score. Why keep them combined? What if certain symptoms or severity improved, but not others following the intervention?

Results:

1. Was there any indication of bias in the survey non-responders? What I mean by this is, I wonder if the individuals who did not complete the surveys were experiencing more/more severe symptoms than the respondents, which could bias your results.

2. I do not think it is appropriate to classify the individuals who reported symptom improvement at 6 weeks and worsening at 12 weeks as "marginal improvement." Honestly, what it sounds like to me is that those individuals misunderstood the question and answered assessing their symptoms compared to the 6 week mark, rather than before vaccination. I would follow up with those participants and clarify that their symptoms were indeed worse after a reported improvement.

3. It is not possible to draw conclusions from your S&S results. For example, you report that symptoms decrease from before vaccination to 2 weeks post-vaccination. While the median does decrease slightly, the IQRs almost completely overlap and the Q3 value actually increases after vaccination. This uncertainty persists throughout the results.

4. I am confused why you report some p-values as adjusted and some as unadjusted. I actually don't think it is particularly appropriate to be reporting statistics on these comparisons, given the very small sample size in this cohort.

If you want to present these results as a descriptive study, I think that would be appropriate in a different journal. Otherwise, there is too much uncertainty and almost certainly bias in these results.

Reviewer #3

(Remarks to the Author)

Overall comments:

The paper claims that COVID-19 vaccination improves health outcomes or results in no change in the majority of vaccine-naïve individuals with Long COVID. It also reports significant elevations in SARS-CoV-2-specific T cells and Spike protein-specific IgG levels post-vaccination. Additionally, specific immune features associated with symptom change after vaccination are identified, suggesting potential predictors of health status post-vaccination. This work contributes novel insights into the poorly characterized area of Long COVID's response to vaccination, particularly regarding symptomatic and immune changes. The clinical implications of this research question are important and of great interest to patients and their treating clinicians, especially as an indirect examination of the viral reservoir hypothesis. The involvement of patient advocacy groups in the study design and recruitment is commendable, enhancing the relevance and patient-centeredness of the research.

The study's focus on vaccine-naïve individuals with Long COVID is both a strength and a limitation. First it allows good data for a true pre/post cohort, although the difficulties in recruitment and that there are very few vaccine naive PASC patients who may be open to vaccination in the future make it less relevant. Subsequently, what may be more relevant now is the impact of COVID-19 vaccine boosters on the illness course. Therefore, as a described in specific points below, I think the authors should consider commenting or framing some of their findings on how studying effect of boosters could build on their work. The paper references existing studies and systematic reviews but distinguishes itself by focusing on vaccine-naïve individuals and employing a comprehensive approach to immune profiling. It is likely to contribute significantly to immunologic understanding of PASC.

Overall though the framing and presentation of their findings is well-written, well-researched, and presented in a very readable and understandable way. For areas of improvement, please consider the following general and more specific points:

General areas for improvement:

- I think the paper should better highlight the difficulty in the recruitment process and that it did not get close to its recruitment goal. Although it is mentioned in statistics and the limitations section, this has large impacts on the generalizability of the study, the relevance of this vaccine naive population, and the limited demographic representation in the cohort that should be addressed head on throughout.

- As described below, the results of secondary outcomes are pooled and reported as median of the entire cohort, but may be important to highlight differences broken down by their overall health response to the vaccine, the primary outcome.

- It's not really commented on in the discussion that the most common PASC symptoms remained relatively stable (fatigue, brain fog). These have been shown to be some of the most common and functionally disabling symptoms of the disease, so the overall improvement and decrease in median symptoms can be largely attributed to decrease in other symptoms. That is important to the clinical implications of these findings, as it would seem to imply that the improvement seen may be through more of the peripheral or secondary symptoms rather than fundamentally change the illness. This is relevant given the lack of control group, as potentially that could reflect the natural illness course as some studies have shown that the ME/CFS symptomatology and dysautonomia persist while the other symptoms wane over time.

Specific areas:

Line 73 Abstract (Minor): consider writing out abbreviation for CNTF as it may not be obvious for most readers that it's ciliary neurotrophic factor

Introduction: Excellent intro, clearly outlines the importance of the clinical question with good supporting studies cited.

Line 158 Data Collection: Was any information surveyed about the participants rationale for pursuing the vaccination collected? That is slightly outside the scope of this paper, but if that data is available may provide some context as to why patients may have delayed primary vaccination series and better characterize this small cohort, especially with the recruitment difficulties encountered.

Line 238 Statistical Analysis: This is the first time in the paper that it is stated "the study was terminated early due to an inability to reach the target." This is a very relevant since the study only recruited 16-32% of its target. I recommend putting this information in the abstract under the methods or results section, and earlier in the text in the "Data Collection" section.

Line 341: The classification or rationale for describing the improvement as "marginal" should be made clearer here. As I interpret this classification of patients who improved then got worse by 12 weeks, would this be better classified as "transient improvement"? Or if their 12 week scores were still better than their baseline despite peaking at 6 weeks, overall the improvement may be temporary and marginal, therefore it would be important to clarify that the trend was toward improvement and that they didn't overall get worse over the 12 weeks.

Line 342 (minor): there is a empty close parentheses typo after "analyses)."

Line 343 Results: This portion of the results section uses the median of the pooled cohort. I do think the overall cohort results are important to report here, but would also recommend providing results if there was a differential in these secondary symptom outcomes stratified by the primary outcome. For example, did the 3 patients who reported worse health overall also demonstrate a decrease in symptoms, or did they increase as you may expect based on their response? Did the no change group also have any significant changes in symptomatology?

Line 415 (minor): It is unclear from this text what is meant by "in concordance." Was the reporting based on standards set forth by this paper, or modeled after that prior study?

Line 459 (minor): SCF should be defined here or at least written out the first time it appears using abbreviation

Line 484 Discussion: I think some precision around the language "vaccination was not overtly harmful" needs some precision or modifiers. For the 3 patients who felt worse that could be considered harmful. Maybe it's just saying not harmful for most patients, or being more precise that there was no measured immune response (TCR, antibody, auto-antibody, or cytokine) associated with worsening symptoms. Furthermore, the discussion does not comment on the 1 patient who was hospitalized for chest pain, which does seem like a very worrisome event in a small cohort.

Line 507 (minor): Is there a citation to support this claim? "Vaccination could also indirectly contribute to the control of latent virus reactivation by restoring proper T and B cell immunity against these herpesviruses"

Line 525: In response to this statement "Elevated interferon signaling suggests the possibility of an ongoing infectious viral process in these individuals," is it possible that elevated IFN could also just be a sign of persistent immune dysregulation not from ongoing infection, but just that this inflammatory was never turned off appropriately?

Line 545 Limitations: I think it is important to highlight that white women made up most of the cohort, which may also factor into the socioeconomic capacity to travel to New Haven and pursue quaternary center research opportunities. The limitations section should address the unrepresentative demographics of this small cohort.

Finally, the limitations section should also comment on the inability to recruit the 50-100 planned patients and what that means given the lack of vaccine naive patients amenable to vaccination to further research. Consequently, it may be helpful to comment on how these findings may be important for future studies evaluating symptoms and immune response to patients getting vaccine boosters. This is a common clinical question for patients seeking advice about vaccination or wanting to know the risks of boosters on their PASC recovery.

Line 552 (minor): for the sentence "The top predictor of participant non-improvement upon vaccination were IFN- β and CNTF, and sIL-6R was found to be a predictor of improvement" would recommend the relative change and directionality of the cytokines (stable elevation of IFN-B and increased sIL-6)

Figure 3: The lines representing "participants trajectory across surveys" make this figure very busy and hard to interpret. I am not sure how to better visualize the data or if different colors could better highlight the differences, but at least on social effects the lines are so close together and on first glance almost seems random that I'm not sure it adds much overall to the paper.

Version 1:

Reviewer comments:

Reviewer #1

(Remarks to the Author)

This is a small but interesting case series, using a very wide range of lab assays and symptom reporting to consider whether vaccination of 16 LC patients who were previously vaccine-naïve leads to any change, for better or worse, in symptom profile. The assays appear to have been designed well and conducted to a high standard.

The previous referee reports and author responses looks to have been a healthy and substantive one - I can see that precision in the manuscript has been sharpened quite a lot. This really matters as the changes are quite nuanced and non-trivial to report accurately, with great potential for the study to be misconstrued/misreported by others. I note the equivocation about how to properly weight glass half-full/half-empty reporting. I think the authors have got this about right - that is, reporting the observations without strong claims one way or the other.

Reviewer #2

(Remarks to the Author)

Reviewer #2 (Remarks to the Author):

Thank you for the opportunity to review the manuscript titled, "Impact of COVID-19 vaccination on symptoms and immune phenotypes in vaccine-naïve individuals with Long COVID." While an interesting premise, based on the cited literature it seems like this question has already been addressed. Unfortunately, I do not see that the present study contributes to the literature, especially as the sample size of this study is too small to even attempt to draw statistical conclusions.

Response: We appreciate this perspective but are unaware of anyone who has addressed this question. The third reviewer was enthusiastic about the rationale for the study. We have helped clarify the gap in knowledge adding the following text to the introduction.

...

REVIEWER RESPONSE: I guess I am confused because you cite 9 papers on vaccination and Long COVID, including several review papers. Is what makes your paper unique that you focused on vaccine naïve individuals (except what about citation #48)? This should be made clear. I will then highlight Reviewer 3's comment: "The study's focus on vaccine-naïve individuals with Long COVID is both a strength and a limitation. ... that there are very few vaccine naïve PASC patients who may be open to vaccination in the future make it less relevant. Subsequently, what may be more relevant now is the impact of COVID-19 vaccine boosters on the illness course." I therefore agree that a re-framing is necessary – perhaps focus more on the immunologic findings as the most interesting contribution, as highlighted by the other two reviewers, rather than on symptom changes and improvement.

Methods:

6. How long before the study were people's infections? That is not mentioned and would impact the prevalence and severity of long COVID symptoms, according to reports in the current literature.

Response: Thank you for raising this point. We have added the following text to the results.

REVIEWER RESPONSE: Sufficiently addressed.

7. There was also no mention of how long people had been experiencing symptoms, and whether they had been improving, stable, or worsening prior to the study's vaccine intervention.

Response: We agree that further describing participants' symptom experience before vaccination is helpful. We do not have information on the trajectory of symptoms before the study, but we can describe how many symptoms per participants had gone away before vaccination.

...

REVIEWER RESPONSE: Sufficiently addressed.

8. There does not appear to be any attempt at having a control group, although there are several comparisons that could have been done. Ideally, you would have a sham vaccine group (or a group who received a vitamin, or a different vaccine, etc.) so that you could compare changes in symptoms and their severity (S&S). You also could have compared S&S to individuals with long COVID who HAD been vaccinated to help understand the impact vaccination might have had on symptom progression.

Response: We agree that a comparison group would be useful in estimating the effect of vaccination on long COVID.

However, at the time of the study in 2021, COVID-19 vaccines were recommended for individuals with long COVID, so randomizing individuals to vaccination or withholding vaccination (whether it was a sham vaccine, vaccine for flu, supplements, etc.) would have been unethical. A concurrent control group of individuals with long COVID who decided not to be vaccinated would have its challenges because individuals who decided not to be vaccinated likely differ systematically from those who decided to be vaccinated (subjecting effect estimates to confounding and selection bias). For these reasons, we adopted the current design where unvaccinated individuals planning to become vaccinated were observed and queried on their long COVID experience before and after vaccination.

REVIEWER RESPONSE: My original comment was not suggesting you should have randomized anything. I was suggesting that you enroll another observational “arm” in your study where you had individuals who refused COVID vaccination all together. I am not sure how these individuals would be systematically different from those you enrolled, who seemed to be incredibly late vaccine recipients anyway; and what would have been different enough to alter their likelihood of developing Long COVID? However, even if you did believe there were systematic differences, you could have measured the appropriate covariates to control for these differences in the analysis. Again, citing Reviewer 3: “That is important to the clinical implications of these findings, as it would seem to imply that the improvement seen may be through more of the peripheral or secondary symptoms rather than fundamentally change the illness. This is relevant given the lack of control group, as potentially that could reflect the natural illness course as some studies have shown that the ME/CFS symptomatology and dysautonomia persist while the other symptoms wane over time.” I believe the lack of a control group critically impacts the conclusions that can be drawn from this paper.

9. Why were TCR and immune markers only quantified in a subset? Who were these individuals?

Response: Only 11 out of 16 participants could be included in the final analysis because of loss to follow-up in biospecimen collection and instances of difficulty in blood draws where less than expected volumes of whole blood were collected. This information has now been included with the text below.

...

REVIEWER RESPONSE: Sufficiently addressed.

10. I am not a fan of combining symptoms and their severity into one score. Why keep them combined? What if certain symptoms or severity improved, but not others following the intervention?

Response: We appreciate this critique and would have liked to pursue more granular analyses on changes in symptom prevalence and duration and symptom severity in terms of the pain and discomfort (i.e., “physical effects”) and impaired social functioning (i.e., “social effects”) associated with each symptom. However, we want to be careful not to overinterpret the data in this small sample. Our goal in combining scores was to provide a single number summary of the overall symptom burden of each participant at each time point (i.e., before vaccination and 2, 6, and 12 weeks post-vaccination), effectively reducing the dimensionality of the data. While we are still careful not to overinterpret granular symptom data, we propose including two supplementary figures and two additional supplementary tables to share more data on symptoms in this cohort (Supplementary Figure 1, Supplementary Figure 2, Supplementary Table 2, Supplementary Table 3)

...

REVIEWER RESPONSE: My initial comment was suggesting that symptoms and severity be assessed separately. They can still be collapsed into a “symptom score” and a “severity score” since I absolutely understand the limitations and concern with over interpreting the data. My concern is that someone could have 10 symptoms, but they could be relatively minor, while someone else could have two symptoms but have them be incredibly severe, and a third could have 10 severe symptoms. And those people are all different. You can disaggregate slightly more than you have and present the findings without requiring deep interpretation. In my opinion having the data out there where it can be evaluated and compared with future studies even without the ability to specifically interpret is more useful than a finding that is hard to actually find meaning in even though it makes the statistics prettier.

11. Was there any indication of bias in the survey non-responders? What I mean by this is, I wonder if the individuals who did not complete the surveys were experiencing more/more severe symptoms than the respondents, which could bias your results.

Response: Thank you for considering this. We want to clarify, here and in the results (line 379), that all participants submitted all four surveys. However, some surveys from three participants were not submitted on time and were removed from the analysis to avoid introducing recall reliability and bias issues.

...

REVIEWER RESPONSE: Thank you for the clarification that all surveys were completed. Can you please define “on time” more specifically in the text? And do you have any reason to suspect that not completing surveys on time could have introduced bias? For example, if I’m participating in a study, I want to do my best to provide all of the information, but I’m not going out of my way when I’m not feeling well to do it. I’m particularly curious about the individuals who reported 3 symptoms and 40 symptoms. The former is an outlier in your data, and the latter is probably the second most symptomatic participant. I’m worried that you’re discarding data that could be informative.

12. I do not think it is appropriate to classify the individuals who reported symptom improvement at 6 weeks and worsening at 12 weeks as "marginal improvement." Honestly, what it sounds like to me is that those individuals misunderstood the question and answered assessing their symptoms compared to the 6 week mark, rather than before vaccination. I would follow up with those participants and clarify that their symptoms were indeed worse after a reported improvement.

Response: We have rephrased "marginal improvement" to "transient improvement" throughout the manuscript, as suggested by Reviewer 3. While participants may have misunderstood questions throughout the survey, we think it is unlikely this is the case for the question asking about overall health (which was used to classify individuals as experiencing "transient improvement" which was phrased, "Compared to your health before you received the first dose of your vaccine on [date], would you say that your health is currently...better...worse...the same...don't know?")

REVIEWER RESPONSE: Sufficiently addressed.

13. It is not possible to draw conclusions from your S&S results. For example, you report that symptoms decrease from before vaccination to 2 weeks post-vaccination. While the median does decrease slightly, the IQRs almost completely overlap and the Q3 value actually increases after vaccination. This uncertainty persists throughout the results.

Response: Thank you for this point. We agree that there is uncertainty in the reported changes in symptom severity given the sample size. Still, an important finding from this data is that there was not a meaningful harm signal. We have added additional supplementary tables (Supplementary Tables 2 and 3 and Supplementary Figure 1 and 2) that we think better characterize the data on symptom severity in this sample, and we remain careful not to overinterpret this additional data.

REVIEWER RESPONSE: Sufficiently addressed.

14. I am confused why you report some p-values as adjusted and some as unadjusted. I actually don't think it is particularly appropriate to be reporting statistics on these comparisons, given the very small sample size in this cohort.

Response: In this study, we tested a few hypotheses and reported p-values. However, given the small sample size, we used stringent multiple-testing correction methods to minimize Type I errors as much as possible. For those comparisons that were not statistically significant before correction, unadjusted p-values were reported. We have removed all the unadjusted p-values from the revised manuscript to avoid confusion.

REVIEWER RESPONSE: You do not mention in your methods anywhere what method you use for your multiple comparisons. In Figure 4's legend, you say that you used the Tukey method, but I assume that is for those data and not the study as a whole. Tukey would not be appropriate for comparing the symptom/severity data, as that test assumes that the data from the different groups come from populations where the observations have a normal distribution and equal standard deviation.

Reviewer #3

(Remarks to the Author)

I would like to extend my appreciation to the authors for their detailed and thoughtful revisions to the manuscript titled "Impact of COVID-19 vaccination on symptoms and immune phenotypes in vaccine-naïve individuals with Long COVID." The authors have made substantial efforts to address the salient points identified in my original review, and their responses have significantly enhanced the quality and clarity of the manuscript.

I commend the authors for the nuanced explanation of the recruitment process and the challenges faced, which adds important context to the study's generalizability. The recharacterization of the "marginal improvement" group to "transient improvement" is a welcome change that better captures the dynamic nature of the participants' health responses. Additionally, the adjustments made to the supplementary figures and tables, which now more effectively illustrate symptom changes among different groups, greatly improve the interpretability of the results.

While this study involves a small cohort, it serves as foundational research on the immunology of vaccine response in Long COVID patients. The detailed immunophenotyping and symptom tracking provide crucial insights that will be instrumental for future research, especially as vaccine boosters become a consistent part of the public health strategy against COVID-19 variants.

Version 2:

Reviewer comments:

Reviewer #2

(Remarks to the Author)

Manuscript Number: COMMSMED-24-0059

Impact of COVID-19 vaccination on symptoms and immune phenotypes in vaccine-naïve individuals with Long COVID

Reviewer #1 (Remarks to the Author):

This manuscript reports a small but interesting study on 16 people with Long Covid in terms of the effect of 1 vaccine dose on symptoms, spike Ab and T cell immunity, a REAP Ab array panel, and Luminex cytokine responses. For the most part, its too small a study for major answers, but there are nevertheless some interesting components.

Much of my reviewer feedback to the authors relates to nuance in terms of how they have elected to report the results:

1. On reaching the Discussion, the authors' message is that the study may allay the vaccine-hesitancy of some with Long Covid fearing that vaccination may exacerbate their symptoms. However, this point - an important one - is never highlighted in the Abstract. Even in the Discussion, a few extra sentences would be helpful in relation to data from Ziyad Al-Aly and others on the benefit of vaccination against ongoing/cumulative LC risk from repeat infections.

Response: Thank you for this comment; this is an important point. Our results are nuanced and the findings are insufficient to allay vaccine hesitancy, even as we show that most people improved. The data from Dr. Al-Aly, who we know well, is useful but not definitive. We prefer not to take a position on the vaccine in this study as the intent is to describe the experience of these individuals and the biological correlates of their outcomes.

2. There's a real glass-half-full or half-empty dilemma in how/whether to report 'symptom improvement': in the 'improved' group, the median number of symptoms had decreased from 23 to 15.5. At first sight, this doesn't seem a huge improvement. Is there any way the authors can better annotate the improvement or otherwise to quality of life? ie was this a substantive improvement?

Response: We appreciate this perspective and agree readers may interpret these data differently. We have said that the strongest conclusion based on the symptoms is that there was no clear signal of improvement or worsening after vaccination. Nevertheless, one person was hospitalized with myocarditis. However, the median number of symptom improvement indicates fewer symptoms after vaccination, and the global health question shows that, generally, more people felt that their health improved.

3. Perhaps the biggest weakness, apart from the lack of an unvaccinated control group, is that the study only proceeds as far as week 12. Since its only 16 people, is there any way of going back to them and asking whether improvement was sustained at 6 or 12 months?

Response: We agree that a longer follow-up would be beneficial to help understand if the changes in symptoms and immunophenotyping after vaccination are sustained or vary. We cannot do further follow-up at this time.

4. Another weakness is that the study encompasses three different vaccines.

Response: Thank you for noting this limitation. We have added the following statement to the Discussion.

(Please note, all line numbers correspond to the line numbers in the manuscript document with tracked changes)

Lines 1038-1040: "Participants procured their vaccinations and received vaccines from three different manufacturers. Potential differences in symptoms and immunophenotype changes between manufacturers were not examined due to the sample size."

5. Sixteen is a very small number of individuals to stratify into improved versus not-improved, but nevertheless, the biosignature they get from the plasma analytes is probably the most newsworthy part of the paper - lines 474-475 - but this didn't come through very clearly in the Abstract or Discussion.

Response: This is an excellent point. We have highlighted this finding in the abstract and the first paragraph of the discussion

Lines 55-57: "Symptom outcomes were most associated with plasma biosignatures. Higher baseline sIL-6R was associated with symptom improvement, and stably elevated levels of IFN- β and CNTF were associated with no improvement"

Lines 932-934: "These outcomes were associated with their plasma-derived biosignatures, suggesting that these immune signatures may serve to differentiate or predict outcomes in future larger studies."

Reviewer #2 (Remarks to the Author):

Thank you for the opportunity to review the manuscript titled, “Impact of COVID-19 vaccination on symptoms and immune phenotypes in vaccine-naïve individuals with Long COVID.” While an interesting premise, based on the cited literature it seems like this question has already been addressed. Unfortunately, I do not see that the present study contributes to the literature, especially as the sample size of this study is too small to even attempt to draw statistical conclusions.

Response: We appreciate this perspective but are unaware of anyone who has addressed this question. The third reviewer was enthusiastic about the rationale for the study. We have helped clarify the gap in knowledge adding the following text to the introduction.

(Please note, all line numbers correspond to the line numbers in the manuscript document with tracked changes)

Lines 110-112: “Consequently, it is crucial to investigate the effect of vaccination on Long COVID symptoms and immunophenotypes by observing individuals before and after their first COVID-19 vaccination, which has remained understudied.”

I have provided specific comments and questions below.

Methods:

6. How long before the study were people’s infections? That is not mentioned and would impact the prevalence and severity of long COVID symptoms, according to reports in the current literature.

Response: Thank you for raising this point. We have added the following text to the results.

Lines 366-369: “The median number of months from participants’ first self-reported positive COVID-19 test until completing the pre-vaccine survey was 5.5 months (Q1-Q3 4.3-12.9, min-max 1.7-19.5). Two participants did not report positive tests, but one reported hospitalization with COVID-19.”

7. There was also no mention of how long people had been experiencing symptoms, and whether they had been improving, stable, or worsening prior to the study’s vaccine intervention.

Response: We agree that further describing participants’ symptom experience before vaccination is helpful. We do not have information on the trajectory of symptoms before the study, but we can describe how many symptoms per participant had gone away before vaccination.

We have added the following text to the results.

Lines 365-366: “The median number of months from participants’ first onset of symptoms until completing the pre-vaccine survey was 7.2 months (Q1-Q3 4.5-13.9, min-max 2.5-19.6).”

Lines 388-391: “The median number of symptoms per participant that had resolved before vaccination was 9 (Q1-Q3 5-15, min-max 1-28), and the most common symptoms experienced that had resolved before vaccination were cough, diarrhea, and persistent chest pain or pressure (n=6 for each).”

8. There does not appear to be any attempt at having a control group, although there are several comparisons that could have been done. Ideally, you would have a sham vaccine group (or a group who received a vitamin, or a different vaccine, etc.) so that you could compare changes in symptoms and their severity (S&S). You also could have compared S&S to individuals with long COVID who HAD been vaccinated to help understand the impact vaccination might have had on symptom progression.

Response: We agree that a comparison group would be useful in estimating the effect of vaccination on long COVID. However, at the time of the study in 2021, COVID-19 vaccines were recommended for individuals with long COVID, so randomizing individuals to vaccination or withholding vaccination (whether it was a sham vaccine, vaccine for flu, supplements, etc.) would have been unethical. A concurrent control group of individuals with long COVID who decided not to be vaccinated would have its challenges because individuals who decided not to be vaccinated likely differ systematically from those who decided to be vaccinated (subjecting effect estimates to confounding and selection bias). For these reasons, we adopted the current design where unvaccinated individuals planning to become vaccinated were observed and queried on their long COVID experience before and after vaccination.

9. Why were TCR and immune markers only quantified in a subset? Who were these individuals?

Response: Only 11 out of 16 participants could be included in the final analysis because of loss to follow-up in biospecimen collection and instances of difficulty in blood draws where less than expected volumes of whole blood were collected. This information has now been included with the text below.

Lines 370-372: “Immunophenotyping assays were completed on a subset of 11 out of 16 participants because of loss to follow-up in biospecimen collection and instances of difficulty in blood draws where less than expected volumes were collected.”

10. I am not a fan of combining symptoms and their severity into one score. Why keep them combined? What if certain symptoms or severity improved, but not others following the intervention?

Response: We appreciate this critique and would have liked to pursue more granular analyses on changes in symptom prevalence and duration and symptom severity in terms of the pain and discomfort (i.e., “physical effects”) and impaired social functioning (i.e., “social effects”) associated with each symptom. However, we want to be careful not to overinterpret the data in this small sample. Our goal in combining scores was to provide a single number summary of the overall symptom burden of each participant at each time point (i.e., before vaccination and 2, 6, and 12 weeks post-vaccination), effectively reducing the dimensionality of the data. While we are still careful not to overinterpret granular symptom data, we propose including two supplementary figures and two additional supplementary tables to share more data on symptoms in this cohort (Supplementary Figure 1, Supplementary Figure 2, Supplementary Table 2, Supplementary Table 3)

The first figure (Supplementary Figure 1) reports boxplots for the number of symptoms per participant that resolved, improved but did not resolve, remained not an issue, remained an issue, or worsened at 2, 6, and 12 weeks post-vaccination compared to pre-vaccination. This is reported for both the physical effects and social effects scales. Data from this table are now described in the results. The second figure (Supplementary Figure 2) is similar to the first but stratifies participants by whether they reported better, the same, or worse overall health at week 12, at the suggestion of Reviewer 3. The text below has been added to the manuscript regarding these figures in the methods and results.

Lines 277-281: “The distribution of the number of symptoms per participant that resolved, improved, remained not an issue, remained an issue, or worsened stayed the same 2, 6, and 12 weeks after vaccination compared to before vaccination was plotted for the physical and social effects scales, overall and by overall health at 12 weeks (better, the same, worse) (symptom changes are defined in the footnotes of Supplementary figure 1).”

Lines 416-440: “The median number of symptoms per participant before vaccination was 23 (Q1-Q3 14.5-27, min-max 3-43, n=16). In terms of pain or discomfort associated with a symptom, the median (Q1-Q3) number of symptoms per participant that resolved at 2, 6, and 12 weeks was 4.5 (3-7.8), 4 (3-7), and 5 (3-12), respectively, and the number of symptoms per participant that improved but did not fully resolve was 6 (6-11), 9.5 (8.3-13.3), and 11 (5-13), respectively (Supplementary figure 1). The median (Q1-Q3) number of symptoms per participant that worsened at 2, 6, and 12 weeks was 3 (2.5-5), 3 (2.5-5), 3 (1-4.3), respectively. Similar trends were observed regarding impairment of social and family functioning associated with a symptom, though with a higher number of symptoms per participant resolving.”

Lines 440-441: “Symptom changes by overall health at 12 weeks is shown in Supplementary figure 2.”

Regarding the last two tables (Supplementary Table 2 and Supplementary Table 3), both provide the number of participants whose symptom—for the fifteen most frequently reported symptoms pre-vaccination—resolved, improved, remained not an issue, remained an issue, or worsened at 2, 6, and 12 weeks post-vaccination compared with pre-vaccination. The first table (Supplementary Table 2) reports symptom severity related to the pain and discomfort of each symptom (i.e., “physical effects”), and the second table (Supplementary Table 3) reports

symptom severity related to impaired social functioning (i.e., “social effects”). The text below has been added to the manuscript regarding these tables in the methods and results.

Lines 281-284: “The proportion of participants experiencing each symptom before vaccination was calculated, and for the fifteen most common symptoms before vaccination, the number of participants whose symptom changed—according to the categories above—was tabulated for the physical and social effects scales.”

Lines 441-443: “For the 15 most common symptoms experienced before vaccination, the number of participants whose symptoms resolved, improved, remained not an issue, remained an issue, or worsened are presented in Supplementary tables 2 and 3.”

11. Was there any indication of bias in the survey non-responders? What I mean by this is, I wonder if the individuals who did not complete the surveys were experiencing more/more severe symptoms than the respondents, which could bias your results.

Response: Thank you for considering this. We want to clarify, here and in the results (line 379), that all participants submitted all four surveys. However, some surveys from three participants were not submitted on time and were removed from the analysis to avoid introducing recall reliability and bias issues.

One participant submitted the 2-week and 6-week surveys around 6 weeks post-vaccination, so their 2-week survey was removed from the analysis. Another participant submitted the 2-week, 6-week, and 12-week surveys around 12 weeks of vaccination, so their 2-week and 6-week surveys were removed from the analysis. Finally, one other participant submitted the 6-week survey after the 2-week survey, so their 6-week survey was removed from the analysis.

We don't think there is a strong indication that these three participants experience more or more severe symptoms than others. For instance, the number of symptoms reported before vaccination among these three participants was 3, 21, and 40 whereas a median of 24 symptoms per participant (Q1-Q3 15-26, min-max 6-43) was reported for the other participants.

We have added the following text to the Introduction and Discussion.

Lines 358-361 (only the words “on time” were added): “Among 429 individuals screened between May 3, 2021 and February 2, 2022, 22 met inclusion criteria and consented to participate and 16 individuals completed the baseline survey and received a first dose of a COVID-19 vaccine; 14 completed the 2-week post-vaccination survey on time, 14 completed the 6-week survey, and 16 completed the 12-week survey.”

Lines 1040-1044: “Surveys from three participants were not completed on time and were excluded from the analysis, but these participants did not seem to have systematically worse or less symptom burden before vaccination; at 12 weeks, one reported overall better health and two reported the same health compared to before vaccination.”

12. I do not think it is appropriate to classify the individuals who reported symptom improvement at 6 weeks and worsening at 12 weeks as “marginal improvement.” Honestly, what it sounds like to me is that those individuals misunderstood the question and answered assessing their symptoms compared to the 6 week mark, rather than before vaccination. I would follow up with those participants and clarify that their symptoms were indeed worse after a reported improvement.

Response: We have rephrased “marginal improvement” to “transient improvement” throughout the manuscript, as suggested by Reviewer 3. While participants may have misunderstood questions throughout the survey, we think it is unlikely this is the case for the question asking about overall health (which was used to classify individuals as experiencing “transient improvement” which was phrased, “Compared to your health before you received the first dose of your vaccine on [date], would you say that your health is currently...better...worse...the same...don’t know?”

13. It is not possible to draw conclusions from your S&S results. For example, you report that symptoms decrease from before vaccination to 2 weeks post-vaccination. While the median does decrease slightly, the IQRs almost completely overlap and the Q3 value actually increases after vaccination. This uncertainty persists throughout the results.

Response: Thank you for this point. We agree that there is uncertainty in the reported changes in symptom severity given the sample size. Still, an important finding from this data is that there was not a meaningful harm signal. We have added additional supplementary tables (Supplementary Tables 2 and 3 and Supplementary Figure 1 and 2) that we think better characterize the data on symptom severity in this sample, and we remain careful not to overinterpret this additional data.

14. I am confused why you report some p-values as adjusted and some as unadjusted. I actually don’t think it is particularly appropriate to be reporting statistics on these comparisons, given the very small sample size in this cohort.

Response: In this study, we tested a few hypotheses and reported p-values. However, given the small sample size, we used stringent multiple-testing correction methods to minimize Type I errors as much as possible. For those comparisons that were not statistically significant before correction, unadjusted p-values were reported. We have removed all the unadjusted p-values from the revised manuscript to avoid confusion.

If you want to present these results as a descriptive study, I think that would be appropriate in a different journal. Otherwise, there is too much uncertainty and almost certainly bias in these results.

Response: We thank the reviewer for their opinion. We respectfully disagree and believe these results are important to share and will have value to readers.

Reviewer #3 (Remarks to the Author):

Overall comments:

The paper claims that COVID-19 vaccination improves health outcomes or results in no change in the majority of vaccine-naïve individuals with Long COVID. It also reports significant elevations in SARS-CoV-2-specific T cells and Spike protein-specific IgG levels post-vaccination. Additionally, specific immune features associated with symptom change after vaccination are identified, suggesting potential predictors of health status post-vaccination. This work contributes novel insights into the poorly characterized area of Long COVID's response to vaccination, particularly regarding symptomatic and immune changes. The clinical implications of this research question are important and of great interest to patients and their treating clinicians, especially as an indirect examination of the viral reservoir hypothesis. The involvement of patient advocacy groups in the study design and recruitment is commendable, enhancing the relevance and patient-centeredness of the research.

The study's focus on vaccine-naïve individuals with Long COVID is both a strength and a limitation. First it allows good data for a true pre/post cohort, although the difficulties in recruitment and that there are very few vaccine naive PASC patients who may be open to vaccination in the future make it less relevant. Subsequently, what may be more relevant now is the impact of COVID-19 vaccine boosters on the illness course. Therefore, as described in specific points below, I think the authors should consider commenting or framing some of their findings on how studying effect of boosters could build on their work. The paper references existing studies and systematic reviews but distinguishes itself by focusing on vaccine-naïve individuals and employing a comprehensive approach to immune profiling. It is likely to contribute significantly to immunologic understanding of PASC.

Overall though the framing and presentation of their findings is well-written, well-researched, and presented in a very readable and understandable way.

Response: We thank the reviewer for these comments.

For areas of improvement, please consider the following general and more specific points:

General areas for improvement:

15. I think the paper should better highlight the difficulty in the recruitment process and that it did not get close to its recruitment goal. Although it is mentioned in statistics and the limitations section, this has large impacts on the generalizability of the study, the relevance of this vaccine naive population, and the limited demographic representation in the cohort that should be addressed head on throughout.

Response: It turned out that it was challenging to recruit people just as they were considering vaccination. Many people who favored vaccination had already been vaccinated, and those who had not were reluctant to join the study. We have developed strategies to do decentralized studies, but we were restricted to our catchment area at the time of this study. We have noted the sample size in the manuscript. Nevertheless, a strength of the study is the volume of data we generated from each participant.

16. As described below, the results of secondary outcomes are pooled and reported as median of the entire cohort, but may be important to highlight differences broken down by their overall health response to the vaccine, the primary outcome.

Response: Thank you for this suggestion. Because only 3 participants reported having the same or worse overall health 12 weeks after vaccination, it's difficult to compare measures of a distribution (e.g., medians) between these small groups. However, as we describe in response to the point just below (#17), we have added supplementary material that present the number of symptoms per participant that resolved, improved, remained not an issue, remained an issue, or worsened after vaccination for the entire cohort (Supplementary Figure 1) and by overall health response to the vaccine (Supplementary Figure 2).

17. It's not really commented on in the discussion that the most common PASC symptoms remained relatively stable (fatigue, brain fog). These have been shown to be some of the most common and functionally disabling symptoms of the disease, so the overall improvement and decrease in median symptoms can be largely attributed to decrease in other symptoms. That is important to the clinical implications of these findings, as it would seem to imply that the improvement seen may be through more of the peripheral or secondary symptoms rather than fundamentally change the illness. This is relevant given the lack of control group, as potentially that could reflect the natural illness course as some studies have shown that the ME/CFS symptomatology and dysautonomia persist while the other symptoms wane over time.

Response: Thank you for this comment. You raise an important point about the distinction between whether a symptom completely resolves or improves. To address this, we have included two additional tables that indicate the number of participants whose symptoms resolved, improved, remained not an issue, remained an issue, or worsened after vaccination for the 15 most reported symptoms before vaccination. We provide tables that categorize these symptom changes based on our measurement of how the symptom affects participants in terms of pain and discomfort (Supplementary Table 2) and how the symptom affects their social functioning (Supplementary Table 3). Many participants reported improvement in symptoms, and fewer reported symptoms that worsened or remained an issue after vaccination.

For example, 13 participants reported brain fog before vaccination. At 12 weeks, 3 participants reported their brain fog was associated with no pain/discomfort (i.e., resolved), 6 said it was improved, 2 said it remained not an issue (i.e., brain fog was associated with no pain/discomfort before and 12 weeks after vaccination), 1 said it remained an issue (i.e., brain fog was associated with the same level of pain/discomfort before and 12 weeks after vaccination), and 1 reported their brain fog had worsened.

18. Line 73 Abstract (Minor): consider writing out abbreviation for CNTF as it may not be obvious for most readers that it's ciliary neurotrophic factor

Response: Thank you, we've added this to the abstract.

19. Introduction: Excellent intro, clearly outlines the importance of the clinical question with good supporting studies cited.

Response: Thank you, we are grateful for this positive comment.

20. Line 158 Data Collection: Was any information surveyed about the participants rationale for pursuing the vaccination collected? That is slightly outside the scope of this paper, but if that data is available may provide some context as to why patients may have delayed primary vaccination series and better characterize this small cohort, especially with the recruitment difficulties encountered.

Response: Understanding the attitudes, beliefs, and intentions of individuals with long COVID who initially defer vaccination and then decide to be vaccinated is certainly important and interesting. This could be especially useful in designing interventions or campaigns to increase vaccine uptake in this population. Unfortunately, we did not collect information on the participants' rationale for pursuing vaccination.

21. Line 238 Statistical Analysis: This is the first time in the paper that it is stated "the study was terminated early due to an inability to reach the target." This is a very relevant since the study only recruited 16-32% of its target. I recommend putting this information in the abstract under the methods or results section, and earlier in the text in the "Data Collection" section.

Reponses: We agree this is important to note. We've added the following text to the Introduction (lines 118-119) and Discussion sections (lines 937-938).

(Please note, all line numbers correspond to the line numbers in the manuscript document with tracked changes)

Introduction (last paragraph): "The study was ended early due to difficulties in recruiting eligible individuals."

Discussion (first paragraph): "Challenges in recruiting individuals who met eligibility criteria limited the sample size to 16 (32% of its target size)."

22. Line 341: The classification or rationale for describing the improvement as "marginal" should be made clearer here. As I interpret this classification of patients who improved then got worse by 12 weeks, would this be better classified as "transient improvement"? Or if their 12 week scores were still better than their baseline despite peaking at 6 weeks, overall the improvement may be temporary and marginal, therefore it would be important to clarify that the trend was toward improvement and that they didn't overall get worse over the 12 weeks.

Response: This is an excellent suggestion. The term “marginal improvement” has been replaced by “transient improvement” in the manuscript and the figures.

23. Line 342 (minor): there is a empty close parentheses typo after "analyses)."

Response: We have removed it from the revised manuscript.

24. Line 343 Results: This portion of the results section uses the median of the pooled cohort. I do think the overall cohort results are important to report here, but would also recommend providing results if there was a differential in these secondary symptom outcomes stratified by the primary outcome. For example, did the 3 patients who reported worse health overall also demonstrate a decrease in symptoms, or did they increase as you may expect based on their response? Did the no change group also have any significant changes in symptomatology?

Response: Thank you for this suggestion. We agree it adds value to describe symptom changes by whether the participant reported better, the same, or worse health 12 weeks after vaccination compared to before vaccination. We have added this data in an additional figure (Supplementary Figure 2). Comparisons between outcome groups are difficult to make because of the small sample. Still, for the 3 participants who reported worse health, it seems they may have reported more symptoms worsening or remaining an issue after vaccination compared to participants who reported “better” health.

25. Line 415 (minor): It is unclear from this text what is meant by "in concordance." Was the reporting based on standards set forth by this paper, or modeled after that prior study?

Response: The paper cited here used the same assay method to study autoantibody dynamics during SARS-CoV2 mRNA vaccination in healthy individuals and those having autoimmune diseases without Long COVID. The term “concordance” has been replaced by “alignment” to avoid confusion.

26. Line 459 (minor): SCF should be defined here or at least written out the first time it appears using abbreviation

Response: Thank you, we have written out SCF as stem cell factor (line 820).

27. Line 484 Discussion: I think some precision around the language "vaccination was not overtly harmful" needs some precision or modifiers. For the 3 patients who felt worse that could be considered harmful. Maybe it's just saying not harmful for most patients, or being more precise that there was no measured immune response (TCR, antibody, auto-antibody, or cytokine) associated with worsening symptoms. Furthermore, the discussion does not comment on the 1 patient who was hospitalized for chest pain, which does seem like a very worrisome event in a small cohort.

Response: We agree there was some response heterogeneity, which we note in the results. However, given the up-and-down nature of the symptoms of people with long COVID, we would

not have expected everyone to have a uniform response. We are reporting no evidence of a meaningful harm signal when looking at the group. We have reported that one person was hospitalized for chest pain and diagnosed with myocarditis three days after the first vaccination and hospitalized again after the second dose. We reviewed this participant's medical records, and it may be that their myocarditis preceded vaccination and was induced by long COVID. Whether there are individuals who can be harmed would require a larger study. We cannot exclude that possibility and note it in the limitations. We have also included more information about the hospitalized participant.

Lines 403-406: "One participant was hospitalized for chest pain and myocarditis three days after receiving their first vaccine dose and again after their second dose. This participant reported being previously hospitalized soon after infection with probable myocarditis."

Lines 929-932 (change underlined): "In this prospective cohort study of 16 vaccination-naïve individuals with Long COVID and significant symptoms at baseline, it was observed that most people improved or stayed the same during follow-up, but some experienced worsening, including one participant who was hospitalized with chest pain and myocarditis."

Line 970: "These factors limit the ability to estimate the benefit or harm caused by vaccination."

28. Line 507 (minor): Is there a citation to support this claim? "Vaccination could also indirectly contribute to the control of latent virus reactivation by restoring proper T and B cell immunity against these herpesviruses"

Response: We include this to signal it is theoretically possible since there is no prior evidence. We have added a relevant citation by Shafiee et al. (Eur J Med Res. 2023).

Lines 983-985: "Vaccination could also theoretically indirectly contribute to controlling latent virus reactivation by restoring proper T and B cell immunity against these herpesviruses."

29. Line 525: In response to this statement "Elevated interferon signaling suggests the possibility of an ongoing infectious viral process in these individuals," is it possible that elevated IFN could also just be a sign of persistent immune dysregulation not from ongoing infection, but just that this inflammatory was never turned off appropriately?

Response: We appreciate the insightful comment from the reviewer. We have modified the language in the discussion to reflect this point. Specifically, we have modified the existing sentence to say,

Lines 1001-1003: "Elevated interferon signaling could suggest the possibility of an ongoing infectious viral process in these individuals or sustained inflammatory condition triggered by the acute infection."

We have also added the following sentence.

Lines 1019-1021: "It is also possible that the elevated IFN signaling pattern may be a sign of persistent immune dysregulation, and not from ongoing infection".

30. Line 545 Limitations: I think it is important to highlight that white women made up most of the cohort, which may also factor into the socioeconomic capacity to travel to New Haven and pursue quaternary center research opportunities. The limitations section should address the unrepresentative demographics of this small cohort.

Response: Thank you for raising this important point. We have added the following text to the discussion section.

Lines 1028-1031 (change underlined): The limitations of this study include the lack of concurrent controls, which we did not include for ethical reasons, the small sample size, and our inability to recruit more participants. Ideally, this study should have been conducted when there was more equipoise about the benefits of vaccination. Moreover, we did not have much diversity in this small cohort."

31. Finally, the limitations section should also comment on the inability to recruit the 50-100 planned patients and what that means given the lack of vaccine naive patients amenable to vaccination to further research. Consequently, it may be helpful to comment on how these findings may be important for future studies evaluating symptoms and immune response to patients getting vaccine boosters. This is a common clinical question for patients seeking advice about vaccination or wanting to know the risks of boosters on their PASC recovery.

Response: We agree that this should be included in the discussion. We appreciate you mentioning it and have added the text as shown above in addressing the previous comment.

32. Line 552 (minor): for the sentence "The top predictor of participant non-improvement upon vaccination were IFN- β and CNTF, and sIL-6R was found to be a predictor of improvement" would recommend the relative change and directionality of the cytokines (stable elevation of IFN-B and increased sIL-6)

Response: To better reflect the data trends for IFN-b, CNTF and sIL-6R, we have modified these sentences to say the following.

Lines 1069-1070: "The top predictors of participant non-improvement upon vaccination were stable elevated levels of IFN- β and CNTF. Elevated levels of sIL-6R were found to be a predictor of improvement"

33. Figure 3: The lines representing "participants trajectory across surveys" make this figure very busy and hard to interpret. I am not sure how to better visualize the data or if different colors could better highlight the differences, but at least on social effects the lines are so close together and on first glance almost seems random that I'm not sure it adds much overall to the paper.

Response: Thank you for this feedback. We have removed the lines connecting points across surveys and simplified the color scheme for Figure 2 (formerly Figure 3).

Manuscript Number: COMMSMED-24-0059

Impact of COVID-19 vaccination on symptoms and immune phenotypes in vaccine-naïve individuals with Long COVID

The most recent responses are colored in red for clarity.

Reviewer #1 (Remarks to the Author):

This is a small but interesting case series, using a very wide range of lab assays and symptom reporting to consider whether vaccination of 16 LC patients who were previously vaccine-naïve leads to any change, for better or worse, in symptom profile. The assays appear to have been designed well and conducted to a high standard.

The previous referee reports and author responses looks to have been a healthy and substantive one - I can see that precision in the manuscript has been sharpened quite a lot. This really matters as the changes are quite nuanced and non-trivial to report accurately, with great potential for the study to be misconstrued/misreported by others. I note the equivocation about how to properly weight glass half-full/half-empty reporting. I think the authors have got this about right - that is, reporting the observations without strong claims one way or the other.

Reviewer #2 (Remarks to the Author):

Thank you for the opportunity to review the manuscript titled, “Impact of COVID-19 vaccination on symptoms and immune phenotypes in vaccine-naïve individuals with Long COVID.” While an interesting premise, based on the cited literature it seems like this question has already been addressed. Unfortunately, I do not see that the present study contributes to the literature, especially as the sample size of this study is too small to even attempt to draw statistical conclusions.

Response: We appreciate this perspective but are unaware of anyone who has addressed this question. The third reviewer was enthusiastic about the rationale for the study. We have helped clarify the gap in knowledge adding the following text to the introduction.

...

REVIEWER RESPONSE: I guess I am confused because you cite 9 papers on vaccination and Long COVID, including several review papers. Is what makes your paper unique that you focused on vaccine naïve individuals (except what about citation #48)? This should be made clear. I will then highlight Reviewer 3’s comment: “The study’s focus on vaccine-naïve individuals with Long COVID is both a strength and a limitation. ... that there are very few vaccine naïve PASC patients who may be open to vaccination in the future make it less relevant. Subsequently, what may be more relevant now is the impact of COVID-19 vaccine boosters on the illness course.” I therefore agree that a re-framing is necessary – perhaps focus more on the

immunologic findings as the most interesting contribution, as highlighted by the other two reviewers, rather than on symptom changes and improvement.

Response: We appreciate this feedback and would like to address it point-by-point.

1. The references cited in the manuscript on vaccination and long COVID mentioned here (refs. 18, 44-48) have only focused on symptom changes or have reviewed the same across studies, except for ref. 19 & 43. In ref. 43, only antibody titers were assayed and found to be negatively correlated with change in long COVID symptoms. So far, there has been only one immunophenotyping study on the impact of vaccination on vaccine naïve individuals with long COVID (ref 19) which profiles immune modulator levels and antibody titers before and after vaccination. Of note, ref. 19 found that vaccination led to significantly decreased levels of several pro-inflammatory plasma factors including sCD40L, GRO- α , MIP-1 α , IL-12p40, G-CSF, M-CSF, IL-1 β , and SCF.
2. Previous studies have also shown a lack of coordination between SARS-CoV-2 specific humoral and antibody responses and T cell exhaustion in long COVID (PMID: 35772216, 38212464). Hence, unlike the previous study (ref 19) we also examined T cell changes. Our approach to understanding the biology after vaccination in people with long COVID also included evaluating T cell responses along with anti-spike antibody responses to probe for correlations.
3. To clarify further, the goals and outcomes for immunophenotyping for this study were:
 - a. Goal: To evaluate if T cell responses upon COVID-19 vaccination aligned with anti-viral antibody responses. Outcome: There was a general concordance in antibody levels and TCR scores following vaccination in people with long COVID. Thus, our study supports coordinated T and antibody responses.
 - b. Goal: To understand if baseline immune signatures could predict symptom changes after vaccination. Outcome: Higher baseline sIL-6R was associated with symptom improvement, and stably elevated levels of IFN- β and CNTF were associated with no improvement. This study is the first to demonstrate baseline biomarkers associated with vaccine outcomes.
 - c. Goal: To identify if alleviation of symptoms upon vaccination resulted in changes in immune modulator levels, which could serve as biomarkers. Outcome: We did not observe significant changes in immunological measurements in any of the outcome groups. Thus, the findings of ref. 19 are not replicated in our study and may point to differences in patient demographics, disease severity or other factors.

We believe, given the diversity in long COVID sequelae, the integrated approach and the unique biosignatures that we report here, add significantly to the findings reported so far. To further emphasize on the immunophenotyping results as suggested by the reviewer, we have revised the abstract conclusion. Finally, while the impact of COVID booster doses on the symptoms and immune phenotypes in long COVID is an area of study that applies to the current situation, our research represents precious and unique data that is impossible to replicate now. It helps understand the nature of the immune responses to the first COVID vaccination in individuals with long COVID. This insight is quite important to the patients, as some believe that long COVID rendered their immune system so defective that vaccinations are useless in boosting immunity against COVID. We show this is not the case. In addition, the results of this study will inform the biomarkers to look for in future studies

examining the impact of booster doses on long COVID symptoms.

Methods:

6. How long before the study were people's infections? That is not mentioned and would impact the prevalence and severity of long COVID symptoms, according to reports in the current literature.

Response: Thank you for raising this point. We have added the following text to the results.

REVIEWER RESPONSE: Sufficiently addressed.

7. There was also no mention of how long people had been experiencing symptoms, and whether they had been improving, stable, or worsening prior to the study's vaccine intervention.

Response: We agree that further describing participants' symptom experience before vaccination is helpful. We do not have information on the trajectory of symptoms before the study, but we can describe how many symptoms per participants had gone away before vaccination.

...

REVIEWER RESPONSE: Sufficiently addressed.

8. There does not appear to be any attempt at having a control group, although there are several comparisons that could have been done. Ideally, you would have a sham vaccine group (or a group who received a vitamin, or a different vaccine, etc.) so that you could compare changes in symptoms and their severity (S&S). You also could have compared S&S to individuals with long COVID who HAD been vaccinated to help understand the impact vaccination might have had on symptom progression.

Response: We agree that a comparison group would be useful in estimating the effect of vaccination on long COVID. However, at the time of the study in 2021, COVID-19 vaccines were recommended for individuals with long COVID, so randomizing individuals to vaccination or withholding vaccination (whether it was a sham vaccine, vaccine for flu, supplements, etc.) would have been unethical. A concurrent control group of individuals with long COVID who decided not to be vaccinated would have its challenges because individuals who decided not to be vaccinated likely differ systematically from those who decided to be vaccinated (subjecting effect estimates to confounding and selection bias). For these reasons, we adopted the current design where unvaccinated individuals planning to become vaccinated were observed and queried on their long COVID experience before and after vaccination.

REVIEWER RESPONSE: My original comment was not suggesting you should have

randomized anything. I was suggesting that you enroll another observational “arm” in your study where you had individuals who refused COVID vaccination all together. I am not sure how these individuals would be systematically different from those you enrolled, who seemed to be incredibly late vaccine recipients anyway; and what would have been different enough to alter their likelihood of developing Long COVID? However, even if you did believe there were systematic differences, you could have measured the appropriate covariates to control for these differences in the analysis. Again, citing Reviewer 3: “That is important to the clinical implications of these findings, as it would seem to imply that the improvement seen may be through more of the peripheral or secondary symptoms rather than fundamentally change the illness. This is relevant given the lack of control group, as potentially that could reflect the natural illness course as some studies have shown that the ME/CFS symptomatology and dysautonomia persist while the other symptoms wane over time.” I believe the lack of a control group critically impacts the conclusions that can be drawn from this paper.

Response: Thank you for clarifying your original comment. We agree with you that studying the causal effect of vaccination on symptoms among people with Long COVID is of great interest. However, such a study would require strong and likely unreasonable assumptions about the comparability of those who chose to be vaccinated and those who did not. Most importantly, it was very difficult to enroll participants, and it would have been even more difficult to match the participants with people who were refusing vaccination. We simply do not have that control group. Nevertheless, our descriptive analysis of data from vaccinated individuals contributes valuable insights and addresses a critical research gap.

We have added this point to the limitations section of the discussion in lines 566-568: “Ideally, this study should have been conducted when there was more equipoise about the benefits of vaccination such that vaccinated participants would be more comparable to unvaccinated individuals with respect to the risk of Long COVID outcomes.”

9. Why were TCR and immune markers only quantified in a subset? Who were these individuals?

Response: Only 11 out of 16 participants could be included in the final analysis because of loss to follow-up in biospecimen collection and instances of difficulty in blood draws where less than expected volumes of whole blood were collected. This information has now been included with the text below.

...

REVIEWER RESPONSE: Sufficiently addressed.

10. I am not a fan of combining symptoms and their severity into one score. Why keep them combined? What if certain symptoms or severity improved, but not others following the intervention?

Response: We appreciate this critique and would have liked to pursue more granular analyses on

changes in symptom prevalence and duration and symptom severity in terms of the pain and discomfort (i.e., “physical effects”) and impaired social functioning (i.e., “social effects”) associated with each symptom. However, we want to be careful not to overinterpret the data in this small sample. Our goal in combining scores was to provide a single number summary of the overall symptom burden of each participant at each time point (i.e., before vaccination and 2, 6, and 12 weeks post-vaccination), effectively reducing the dimensionality of the data. While we are still careful not to overinterpret granular symptom data, we propose including two supplementary figures and two additional supplementary tables to share more data on symptoms in this cohort (Supplementary Figure 1, Supplementary Figure 2, Supplementary Table 2, Supplementary Table 3)

...

REVIEWER RESPONSE: My initial comment was suggesting that symptoms and severity be assessed separately. They can still be collapsed into a “symptom score” and a “severity score” since I absolutely understand the limitations and concern with over interpreting the data. My concern is that someone could have 10 symptoms, but they could be relatively minor, while someone else could have two symptoms but have them be incredibly severe, and a third could have 10 severe symptoms. And those people are all different. You can disaggregate slightly more than you have and present the findings without requiring deep interpretation. In my opinion having the data out there where it can be evaluated and compared with future studies even without the ability to specifically interpret is more useful than a finding that is hard to actually find meaning in even though it makes the statistics prettier.

Response: Thank you for this clarification. We agree that reporting symptoms and their severity is a complex task. The purpose of providing Supplementary Figures 1 and 2 and Supplementary Tables 2 and 3 is to provide a breakdown of how and how many symptoms change over time. While a single-number summary of participant symptom burden cannot disentangle participants with many mild symptoms from participants with few severe symptoms, it can still be a useful marker of overall disease burden. To your point, we have added the following text to the results and Supplementary Table that reports the median number of symptoms per participant that affected participants to varying degrees over time on the physical and social effect measures.

Lines 369-371: “The median number of symptoms that bothered participants “very much”, “quite a bit”, “somewhat”, “a little bit”, or “not at all” before and after vaccination are provided in Supplementary table 4.”

Supplementary Table 4: Median number of symptoms per participant reported as bothering participants “very much”, “quite a bit”, “somewhat”, “a little bit”, or “not at all” on the physical and social effects scales.

Weeks since first COVID-19 vaccine dose	n	Very much	Quite a bit	Somewhat	A little bit	Not at all
Physical effect scale						

Pre-vaccination	16	8 (4-12)	6 (5-9.5)	5 (3.5-7)	3 (1.25-4.75)	4.5 (2.75-6.25)
2 weeks	14	4.5 (1.75-9.25)	6 (3-9)	6 (3-7)	4 (3-6.75)	4 (3-7)
6 weeks	14	3 (3-5)	5 (2.25-6)	4 (3-6)	6 (5-8)	4 (3-7)
12 weeks	16	1.5 (1-3.25)	4.5 (3-5.25)	7 (1-11)	7 (3-9)	6.5 (3.25-11)
Social effect scale						
Pre-vaccination	16	5 (2-7)	3 (2-5.75)	5 (3-6.75)	3 (1-7)	5.5 (4-15.5)
2 weeks	14	1 (1-5)	2.5 (1.75-4.25)	4 (1-5)	5 (3-8.5)	11.5 (6.25-18.25)
6 weeks	14	5 (2-5)	3 (1.5-7)	5 (5-5)	4 (2-9)	8 (4.5-15.75)
12 weeks	16	3 (1-3)	3 (1-5)	3 (2.25-4.75)	6 (2-8)	13 (6.5-21)
Interquartile ranges are shown in parentheses. Physical effects were measured with the question “While experiencing these symptoms, how much do/did they bother you in terms of discomfort or pain?”, and social effects were measured with the question, “After quarantine, how much does/did the symptom impair your social or family functioning compared to pre-COVID?” in the 2-, 6-, and 12-week surveys.						

11. Was there any indication of bias in the survey non-responders? What I mean by this is, I wonder if the individuals who did not complete the surveys were experiencing more/more severe symptoms than the respondents, which could bias your results.

Response: Thank you for considering this. We want to clarify, here and in the results (line 379), that all participants submitted all four surveys. However, some surveys from three participants were not submitted on time and were removed from the analysis to avoid introducing recall reliability and bias issues.

...

REVIEWER RESPONSE: Thank you for the clarification that all surveys were completed. Can you please define “on time” more specifically in the text? And do you have any reason to suspect that not completing surveys on time could have introduced bias? For example, if I’m participating in a study, I want to do my best to provide all of the information, but I’m not going out of my way when I’m not feeling well to do it. I’m particularly curious about the individuals who reported 3 symptoms and 40 symptoms. The former is an outlier in your data, and the latter is probably the second most symptomatic participant. I’m worried that you’re discarding data that could be informative.

Thank you for raising this point. We certainly do not want to discount data and participants’ experiences that we think are valuable and accurate. The scenario you describe is possible, but we simply do not have data to know the participants’ true symptom experience at the survey time points we excluded. Because the study’s goal was to look at changes in symptoms and overall health over time, it is important these data are correctly temporally ordered. Most importantly, we do have complete data for all 16 participants in the final 12-week survey, which is the most

meaningful time point in this study. For this survey, we looked at the number of symptoms reported comparing the 3 participants with excluded surveys to the other 13 participants. We did not observe any clear differences. Please see the table at the end of this response for this information.

For the three participants with excluded surveys, the table below lists the time of their surveys and the number of symptoms that bothered them quite a bit or very much. Bolded cells indicate the excluded surveys. As you can see, it is hard to infer much from these symptoms measured out of the window.

Number of symptoms that bothered participants "quite a bit" or "very much" on the physical effects scale.				
	Pre-vaccination	2-weeks	6-weeks	12-weeks
Participant 1	33	25	11^a	7
Participant 2	11	12^b	9	7
Participant 3	3	3^c	2^c	2

^a Excluded because survey was submitted before the 2-week survey
^b Excluded because survey was submitted on the same day as 6-week survey, around 6 weeks after vaccination.
^c Excluded because surveys were submitted on the same day as the 12-week survey, which was submitted around 12 weeks after vaccination.

We have clarified what we mean by “on time” in the text in Lines 302-307: “Among 429 individuals screened between May 3, 2021 and February 2, 2022, 22 met inclusion criteria and consented to participate and 16 individuals completed the baseline survey and received a first dose of a COVID-19 vaccine. Two 2-week surveys and two 6-week surveys were excluded because they were submitted before an earlier survey time point or on the same day as another survey; thus, we included 14 2-week surveys, 14 6-week surveys, and 16 12-week surveys.”

We address the concern you raise in the Discussion section, Lines 580-584: “Surveys from three participants were excluded from the analysis because their submissions were out of order or on the same day. Among these participants at 12 weeks, one reported overall better health and two reported the same health compared to before vaccination. It’s possible the timing of survey submissions was related to disease severity, for instance delaying study surveys until symptoms subsided, but this unknown.”

Number of symptoms reported in the 12-week survey, by scale (physical, social), severity (Not at all, ..., Very much), and Response Status (i.e., "non-responders" are the three participants where their 2- or 6-week survey was excluded)				
Participant group	Symptom severity (how much it bothered a participant)	Mean	1st quartile	3rd quartile
Physical effect scale				

non-responder	Not at all	4.5	2.8	6.3
responder	Not at all	8.4	3.8	12.5
non-responder	A little bit	8.5	7.8	9.3
responder	A little bit	6.4	3.0	8.5
non-responder	Somewhat	11.0	9.0	13.0
responder	Somewhat	5.9	1.0	9.5
non-responder	Quite a bit	4.3	3.5	5.5
responder	Quite a bit	4.6	3.0	5.0
non-responder	Very much	1.5	1.3	1.8
responder	Very much	2.8	1.0	3.8
Social effect scale				
non-responder	Not at all	13.0	4.5	19.0
responder	Not at all	14.1	6.8	20.5
non-responder	A little bit	4.0	3.0	5.0
responder	A little bit	6.2	2.0	9.0
non-responder	Somewhat	2.7	1.5	3.5
responder	Somewhat	5.4	3.0	7.0
non-responder	Quite a bit	3.0	2.0	4.0
responder	Quite a bit	4.0	1.0	5.0
non-responder	Very much	1.0	1.0	1.0
responder	Very much	3.0	2.5	3.5

12. I do not think it is appropriate to classify the individuals who reported symptom improvement at 6 weeks and worsening at 12 weeks as “marginal improvement.” Honestly, what it sounds like to me is that those individuals misunderstood the question and answered assessing their symptoms compared to the 6 week mark, rather than before vaccination. I would follow up with those participants and clarify that their symptoms were indeed worse after a reported improvement.

Response: We have rephrased “marginal improvement” to “transient improvement” throughout the manuscript, as suggested by Reviewer 3. While participants may have misunderstood questions throughout the survey, we think it is unlikely this is the case for the question asking about overall health (which was used to classify individuals as experiencing “transient improvement” which was phrased, “Compared to your health before you received the first dose of your vaccine on [date], would you say that your health is currently...better...worse...the same...don’t know?”

REVIEWER RESPONSE: Sufficiently addressed.

13. It is not possible to draw conclusions from your S&S results. For example, you report that symptoms decrease from before vaccination to 2 weeks post-vaccination. While the median does decrease slightly, the IQRs almost completely overlap and the Q3 value actually increases after vaccination. This uncertainty persists throughout the results.

Response: Thank you for this point. We agree that there is uncertainty in the reported changes in symptom severity given the sample size. Still, an important finding from this data is that there was not a meaningful harm signal. We have added additional supplementary tables (Supplementary Tables 2 and 3 and Supplementary Figure 1 and 2) that we think better characterize the data on symptom severity in this sample, and we remain careful not to overinterpret this additional data.

REVIEWER RESPONSE: Sufficiently addressed.

14. I am confused why you report some p-values as adjusted and some as unadjusted. I actually don't think it is particularly appropriate to be reporting statistics on these comparisons, given the very small sample size in this cohort.

Response: In this study, we tested a few hypotheses and reported p-values. However, given the small sample size, we used stringent multiple-testing correction methods to minimize Type I errors as much as possible. For those comparisons that were not statistically significant before correction, unadjusted p-values were reported. We have removed all the unadjusted p-values from the revised manuscript to avoid confusion.

REVIEWER RESPONSE: You do not mention in your methods anywhere what method you use for your multiple comparisons. In Figure 4's legend, you say that you used the Tukey method, but I assume that is for those data and not the study as a whole. Tukey would not be appropriate for comparing the symptom/severity data, as that test assumes that the data from the different groups come from populations where the observations have a normal distribution and equal standard deviation.

Response: We have used the Bonferroni correction method to account for multiple comparisons in our analyses of SARS-CoV-2 specific T cell responses and anti-SARS-CoV-2 antibody responses. This is described in the subsection on statistical methods between lines 260-261, as well as in the legend for Figure 3 (line 777). We have added further information about Tukey adjustment to the Methods section for Figure 4 to provide clarity.

The methods now say (lines 269-278): "...to estimate the average differences in expression of each cytokine plasma factor over the course of vaccination (Figure 4), we used linear mixed models via Restricted Maximum Likelihood (REML) regression, estimating the cytokine expression over all three time points amongst three symptom outcome groups: those who did not improve or felt worse at weeks 6 and 12 post-vaccination (n=3; Same/Worse), those who showed transient improvement (n=2, Transient [i.e. Better week 6; then Worse week 12]) and those who reported improvement (n=7, Better). The model incorporated a random effect for each individual as a random intercept, nested within their respective symptom outcome groups. The fixed effects in the model included the symptom outcome and time, along with an interaction term between

them to investigate any potential modifying effect of time on the symptom outcome group and adjusted for multiple comparison within each group for these plasma factors using the Tukey method.

Reviewer #3 (Remarks to the Author):

I would like to extend my appreciation to the authors for their detailed and thoughtful revisions to the manuscript titled "Impact of COVID-19 vaccination on symptoms and immune phenotypes in vaccine-naïve individuals with Long COVID." The authors have made substantial efforts to address the salient points identified in my original review, and their responses have significantly enhanced the quality and clarity of the manuscript.

I commend the authors for the nuanced explanation of the recruitment process and the challenges faced, which adds important context to the study's generalizability. The recharacterization of the "marginal improvement" group to "transient improvement" is a welcome change that better captures the dynamic nature of the participants' health responses. Additionally, the adjustments made to the supplementary figures and tables, which now more effectively illustrate symptom changes among different groups, greatly improve the interpretability of the results.

While this study involves a small cohort, it serves as foundational research on the immunology of vaccine response in Long COVID patients. The detailed immunophenotyping and symptom tracking provide crucial insights that will be instrumental for future research, especially as vaccine boosters become a consistent part of the public health strategy against COVID-19 variants.